# PharmaVQA: A Retrieval-Augmented Visual Question Answering Framework for Molecular Representation via Pharmacophore Guided Prompts

## Abstract

In drug discovery, molecular representation learning is vital for understanding and generating new drug-like molecules. The accurate representation of molecules facilitates drug candidate screening and the optimization of lead compounds. The vastness of chemical space challenges traditional drug design and relies on complex computations. The Pharmacophore is a functional group contained within a drug molecule, which binds to receptors or biological macromolecules to produce biological effects and reduce computations. Pharmacophore-guided representation of molecules, however, remains a significant challenge. To address this issue, we propose an improved deep learning-based model called PharmaVQA for retrieving pharmacophore-related information directly from molecule databases, allowing for a more targeted understanding of drug-like molecules. Through the use of Visual Question Answering (VQA) framework, PharmaVQA captures pharmacophore data, generates knowledge prompts, and enriches molecular representations. On 46 benchmark datasets, PharmaVQA has demonstrated superior performance in both molecular property prediction and drug-target interaction prediction. Additionally, the applicability of PharmaVQA in drug discovery has been validated on an FDA-approved molecule dataset, where the Top-20 predictions were analyzed in real-world studies, with the majority of them experimentally validated as potential ligands previously reported in the literature. Our assessment of PharmaVQA is that it is a powerful and useful tool for accelerating the development of AI-assisted drug discovery across a wide range of areas.

## 1 Introduction

Identifying molecules with specific properties remains challenging in drug discovery due to the time and resources required for experimental validation (Dickson & Gagnon, 2004; Mullard, 2014). AI-driven methods have recently enhanced the efficiency of molecular property prediction, however, developing effective molecular representations remains challenging (Hessler & Baringhaus, 2018; Walters & Barzilay, 2020). Early machine learning-based methods relied on manually crafted features, such as molecule descriptors and FingerPrints (FP), which required complex engineering and limited adaptability (Van De Waterbeemd & Gifford, 2003; Dong et al., 2018; Butler et al., 2018; Li et al., 2023b). Deep learning models like Convolutional Neural Networks (CNNs), Recurrent Neural Networks (RNNs) and Graph Neural Networks (GNNs) have automated feature extraction from Simplified Molecular Input Line Entry System (SMILES) or molecule graphs (Li et al., 2023a; Xu et al., 2017; Shi et al., 2019; Gilmer et al., 2017), but scarcity of labeled data and vast chemical space continue to limit accuracy (Dong et al., 2018; Hu* et al., 2020). Self-supervised learning has improved GNNs via pre-training, but these models struggle to capture detailed molecular semantics, especially long-range interactions (You et al., 2020; 2021), and often fail to capture critical spatial and functional relationships limiting the scalability of complex tasks (Sun et al., 2022; Zhang et al., 2021).

To address these representation challenges, pharmacophores have emerged as a critical concept in drug design (Jiang et al., 2023). Pharmacophores represent the spatial arrangement of functional

groups essential for biological activity, offering a robust framework for understanding molecular interactions with biological targets (Jiang et al., 2023). By identifying the essential features that contribute to binding affinity and pharmacological effects, pharmacophores help simplify molecular representations (Li et al., 2022b). This approach can reduce computational costs by focusing on the key components of a molecule responsible for interactions with biological receptors or macromolecules, thereby improving the efficiency of computational models (Noor et al., 2023). Despite these advantages, representing molecules through a pharmacophore-guided approach remains challenging, primarily due to the difficulty of accurately capturing the diverse arrangements of functional groups that are essential for specific interactions.

Recently, Visual Question Answering (VQA) has been proposed to provide accurate answers to visual and language-based queries (Antol et al., 2015; Ma et al., 2024). The technology harnesses the synergy of Computer Vision (CV) and Natural Language Processing (NLP) methods, enabling a deeper understanding of both image content and textual inquiries. In the field of VQA research, advanced multimodal fusion methodologies have been explored, notably including the Bilinear Attention Network (BAN) (Kim et al., 2018; Guo et al., 2023). BAN selectively attends to salient image features, while adeptly filtering out irrelevant information, thereby enhancing VQA responses precision. This advancement underscores the progress made in the field toward more accurate and nuanced interpretations of complex visual-linguistic queries.

In this study, we introduce the PharmaVQA model (see Figure. 1), a retrieval-based approach that enhances molecular representation by directly retrieving pharmacophore-related information. To optimize pharmacophore knowledge retrieval, we employ VQA technology to construct prompts (queries) for the retrieval process. Although typically used for answering questions based on images, VQA is innovatively applied here to generate knowledge prompts related to molecule properties. By designing appropriate questions, the model can automatically retrieve and integrate answers from multiple sources, forming a comprehensive description of molecule characteristics. In this way, molecular representations are enriched and high-quality data is provided for modeling and applications. We conducted extensive experiments on multiple benchmark datasets related to molecular representation, demonstrating superior performance over existing methods. This validates the effectiveness and generalizability of the proposed approach. The viability of PharmaVQA is demonstrated by identifying potential ligands.

Our contributions are summarized as follows:

- We introduce a novel framework named pharmaVQA, a retrieval-augmented visual question-answering framework that extracts information related to pharmacophores from molecules as a knowledge prompt to enhance molecular features.
- We apply bilinear attention to integrating information from both the question and molecular graph, providing an attention map that enhances PharmaVQA's interpretability.
- We conduct experiments on 46 molecular representation datasets, demonstrating results that surpass existing state-of-the-art (SOTA) methods. PharmaVQA's practical applicability in drug discovery has been validated on three ligand datasets (HPK1, FGFR1, and VIM-1) with 10, 15, and 16 of the Top-20 predictions experimentally confirmed as potential ligands.

## 2 RELATED WORK

**Molecular Representation Learning.** Methods focusing on molecular representation extract molecule graphs that encompass both node (atom or motif) and edge (bond) information. Such as Xia et al. (2023) proposed MoleBERT as a novel masking strategy at the node level and a triplet masked contrastive learning approach at the graph level to achieve a comprehensive view of molecular representations. Recent research focuses on multimodal approaches that combine textual descriptions of molecules with graph representations. MoleculeSTM (Liu et al., 2023) utilizes a contrastive learning strategy to simultaneously learn the chemical structures and textual descriptions of molecules, facilitating a more nuanced comprehension of their dual representations. SPMM (Chang & Ye, 2024) developed a multimodal molecular pre-training model that incorporates both structural and biochemical property modalities. By aligning structural and property features in a shared embedding space, this model captures bidirectional information between molecule structures and

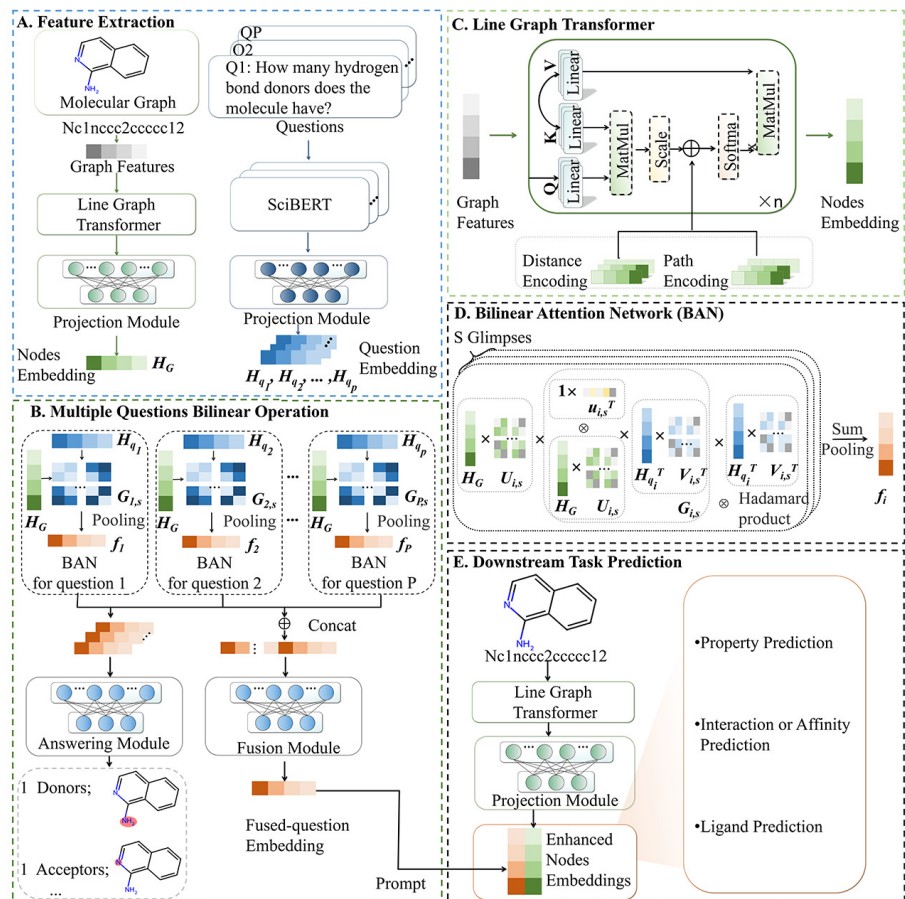

Figure 1: **Overview of PharmaVQA. A** Feature extraction: a line graph transformer and SciBERT model process the molecular graph and question embeddings. **B** Multiple questions bilinear operation: Bilinear Attention Network (BAN) handles multiple question embeddings through bilinear operations, followed by pooling and concatenation. The fused-question embedding integrates into the output. **C** The Line Graph Transformer. **D** The Bilinear Attention Network. **E** Downstream Task Prediction: The fused embeddings are used for various downstream tasks, such as property prediction, interaction or affinity prediction, and ligand prediction, leveraging enhanced node embeddings.

their attributes, enhancing the understanding of their complex relationships. For knowledge-based molecular representation learning methods, KPGT (Li et al., 2023a) employs a pretraining strategy that involves masking a subset of nodes and incorporating a knowledge node to enhance representation, utilizing the LineGraphTransformer to extract node features. However, these methods often overlook the critical spatial and functional relationships, limiting the scalability of complex tasks.

## 3 PRELIMINARY

### 3.1 BILINEAR ATTENTION NETWORK

The Bilinear Attention Network (BAN) model generates an attention map $\boldsymbol{G}$. Subsequently, this attention map $\boldsymbol{G}$ is used, along with visual and textual representations, to produce a combined output vector $\boldsymbol{z}$ that includes information from both modalities. Lastly, the joint embedding $\boldsymbol{z}$ is forwarded to an MLP classifier to assess the answers.

$$\boldsymbol{G} = softmax\left(\left(\left(\mathbf{1} \cdot \boldsymbol{p}^\top\right) \circ \sigma\left(\boldsymbol{Q}^\top \boldsymbol{W}_Q\right)\right) \sigma\left(\boldsymbol{W}_V^\top \boldsymbol{V}\right)\right), \tag{1a}$$

$$\boldsymbol{z} = \sigma\left(\boldsymbol{Q}^\top \boldsymbol{W}_U\right) \circ \boldsymbol{G}\sigma\left(\boldsymbol{W}_V^\top \boldsymbol{V}\right)\mathbf{1}, \tag{1b}$$

$$y = MLP(\boldsymbol{z}), \tag{1c}$$

where $y$ denotes the response, which can be in the form of a string or numerical number of regression or classification problems.

## 3.2 GRAPH TRANSFORMER

Graph Transformer introduces the topological properties of the graph into the Transformer model, enabling the model to handle high-dimensional spatial data with structural position priors, leading to its extensive application across various domains (Yun et al., 2019; Mialon et al., 2021; Wu et al., 2023; Chen et al., 2024). Given a molecular graph $\mathcal{G}$, the initial feature is $\boldsymbol{X} \in \mathbb{R}^{n \times d}$, where $d$ denotes the dimension of atom feature covering the atomic number, atomic type, etc. Utilizing the connectivity relationships between atom nodes, an adjacency matrix $\boldsymbol{A} \in \mathbb{R}^{n \times n}$ can be constructed, where $\boldsymbol{A}_{ij}$ signifies whether node $i$ is connected to node $j$.

$$\hat{\boldsymbol{H}}^{(l)} = \left( \frac{\boldsymbol{H}^{(l-1)} \boldsymbol{W}_Q \times (\boldsymbol{H}^{(l-1)} \boldsymbol{W}_K)^\top}{\sqrt{d}} + \boldsymbol{A} \right) (\boldsymbol{H}^{(l-1)} \boldsymbol{W}_V), \tag{2a}$$

$$\boldsymbol{H}^{(l)} = Residual(\boldsymbol{H}^{(l)}, \hat{\boldsymbol{H}}^{(l)}), \tag{2b}$$

where $\boldsymbol{H}^{(l-1)}$ denotes the node feature matrix on layer $(l-1)$ and $\boldsymbol{H}^{(0)} = \boldsymbol{X}$, $\boldsymbol{W}^Q \in \mathbb{R}^{d \times d_a}$, $\boldsymbol{W}^K \in \mathbb{R}^{d \times d_a}$, $\boldsymbol{W}^V \in \mathbb{R}^{d \times d_a}$ are the trainable parameter matrix which $d_a$ is the dimension of attention module, the $Residual(\cdot)$ is a function to alleviate the problem of gradient vanishing or exploding that may occur during the training process of deep neural networks, and helping the model better learn complex graph-structured data.

## 4 PROPOSED FRAMEWORK: PHARMAVQA

### 4.1 MOLECULAR REPRESENTATION

To incorporate molecular edge information, we perform a pre-processing step (Li et al., 2023a) on the original graph, transforming it into the augmented graph. In this new graph, the node set comprises information about the current edges augmented with the details of the atomic nodes they connect, while the edge set represents the new edges connecting these enriched nodes. We apply the LineGraphTransformer (Li et al., 2023a), referred to as $Encoder_g(\cdot)$, to encode the graph. This transformer utilizes the GraphTransformer for encoding the molecular graph, integrating both the positional encoding and distance encoding modules. As a result, we obtain the representation $\boldsymbol{H}_{\mathcal{G}} \in \mathbb{R}^{n \times d_g}$ for the modified molecular graph $\mathcal{G}$:

$$\boldsymbol{H}_{\mathcal{G}} = Encoder_g(\mathcal{G}), \tag{3}$$

where $d_g$ denotes the dimension of the graph node feature.

### 4.2 QUESTION REPRESENTATION

We design a series of query questions denoted as $\mathbb{Q} = \{\boldsymbol{q}_1, \cdots, \boldsymbol{q}_P\}$, where each pharmacophore question is tailored to uncover the pharmacophore features of the molecule (The details can be found in Section. 5.2 and Appendix A). These queries are then embedded through pre-trained language models such as SciBERT (Beltagy et al., 2019), which capture semantic information within the text. SciBERT's proficiency in understanding scientific text provides robust support for extracting relevant knowledge associated with the pharmacophores, enabling a deeper and more nuanced understanding of their various facets. Suppose the question text for pharmacophore type $i$ is defined as $\boldsymbol{q}_i = [q_i^1, \cdots, q_i^l]$, where $l$ denotes the sequence length. The embeddings corresponding to the text $\boldsymbol{H}_{\boldsymbol{q}_i} \in \mathbb{R}^{l \times d_l}$ is defined as follows:

$$\boldsymbol{H}_{\boldsymbol{q}_i} = Encoder_t([q_i^1, \cdots, q_i^l]), \tag{4}$$

where $Encoder_t(\cdot)$ denotes the SciBERT encoder and $d_l$ denotes the dimension of input text.

### 4.3 PHARMACOPHORE KNOWLEDGE EXTRACTION

To generate predictive outputs that match the questions, we combine the features of the graph $\boldsymbol{H}_{\mathcal{G}}$ and the question text $\boldsymbol{H}_{\boldsymbol{q}}$ through BAN to extract knowledge. Specifically, we produce several attention maps customized for pharmacophore-related queries. In this context, we utilize the multiple

glimpse approach (Kim et al., 2018; Guo et al., 2023) to improve the model's understanding. Given a pharmacophore-related question of type $i$ as $\boldsymbol{q}_i$, the corresponding attention map $\boldsymbol{G}_{i,s} \in \mathbb{R}^{n \times l}$ for the $s$th glimpse representation is as follows:

$$\boldsymbol{G}_{i,s} = softmax\left(\left(\left(\mathbf{1} \cdot \boldsymbol{u}_{i,s}^{\top}\right) \circ \sigma\left(\boldsymbol{H}_{\mathcal{G}} \boldsymbol{U}_{i,s}\right)\right) \sigma\left(\left(\boldsymbol{V}_{i,s} \boldsymbol{H}_{\boldsymbol{q}_i}\right)^{\top}\right)\right), i = 1, \cdots, P, s = 1, \cdots, S,$$
(5)

where $P$ and $S$ denote the overall count of all types of pharmacophore and the number of glimpses. $\mathbf{1} \in R^n$ stands for ones vector, $\boldsymbol{U}_{i,s} \in \mathbb{R}^{d_g \times d_k}$, $\boldsymbol{V}_{i,s} \in \mathbb{R}^{d_l \times d_k}$ and $\boldsymbol{u}_{i,s} \in \mathbb{R}^{d_k}$ are the learnable parameters, $d_k$ represents the dimension of these parameters, $\sigma$ signifies the ReLU activation function, while $\circ$ and $softmax(\cdot)$ represent Hadamard product and softmax function, respectively.

Next, we create joint embeddings $\boldsymbol{f}_{i,s} = [f_{i,s,1}, \cdots, f_{i,s,K}] \in \mathbb{R}^K$ for each unique attention map:

$$f_{i,s,k} = \sigma\left(\boldsymbol{H}_{\mathcal{G}} \boldsymbol{U}_{i,s}\right)_k^{\top} \boldsymbol{G}_{i,s} \sigma\left(\left(\boldsymbol{V}_{i,s} \boldsymbol{H}_{\boldsymbol{q}_i}\right)^{\top}\right)_k, k = 1, \cdots, K.$$
(6)

Subsequently, a sum pooling function (denoted as $SumPool(\cdot)$) is employed to combine the obtained $S$ glimpse vectors, resulting in the features relevant to the pharmacophore $i$ query:

$$\boldsymbol{f}_i = SumPool(\boldsymbol{f}_{i,s}).$$
(7)

Finally, we merge the embeddings designed for various pharmacophore-related questions to combine their unique features. This unified representation, enriched with information from multiple embeddings of pharmacophore questions, is then used for predictions in subsequent tasks.

$$\boldsymbol{f} = MLP(\boldsymbol{f}_1, \cdots, \boldsymbol{f}_P).$$
(8)

### 4.4 PROMPT-BASED PREDICTION TASK

In the subsequent prediction tasks, we integrate the extracted pharmacophore knowledge features $\boldsymbol{f}$ as prompts into the molecular embeddings from another encoder which is derived from $\boldsymbol{H}_{\mathcal{G}}' = Encoder_g'(\mathcal{G})$. This enhanced embedding, which includes both molecular features and pharmacophore-related knowledge, is then utilized for predictions in downstream tasks.

$$\hat{y} = MLP(concat(\boldsymbol{f}, \boldsymbol{H}_{\mathcal{G}}')),$$
(9)

where the $concat(\cdot)$ denotes the cat operation of two vectors.

### 4.5 MODEL TRAINING

For the molecular property prediction task, our model employs different loss functions $L_p$ tailored to a specific task. For the classification task, we adopt the Binary Cross-Entropy (BCE) loss function, which is well-suited to handling binary or multi-class classification problems. On the other hand, for regression tasks, we utilize the Mean Squared Error (MSE) loss function, as it provides a straightforward measure of the difference between the predicted and true values.

Furthermore, for predicting answers corresponding to specific questions, we also employ the MSE regression loss function $L_{ph}$, ensuring that our model is optimized to accurately predict continuous values in the context of question answering.

$$L_{ph} = \frac{1}{N} \sum_{j=1}^{N} \frac{1}{P} \sum_{i=1}^{P} \left(r_i^j - MLP(\boldsymbol{f}_i^j)\right)^2,$$
(10)

where $N$ denotes the total molecule count, $r_i^j$ and $\boldsymbol{f}_i^j$ indicate the true label and the pharmacophore feature vector for the $i$th functional group of the $j$th molecule, and an MLP layer is used to obtain the predicted value $MLP(\boldsymbol{f}_i^j)$. In addition to these task-specific losses, we incorporate an alignment loss $L_{align}$ derived from the Bilinear Attention Network.

In this context, we create a matrix $\boldsymbol{O} \in \mathbb{R}^{n,P}$, where $n$ is the number of nodes in a molecular graph and $P$ represents different functional group questions. Each element in $\boldsymbol{O}$ indicates if a node belongs to the $i$th functional group. By summing the columns of the final glimpse attention map

$\boldsymbol{G}_{i,P}$, we obtain a vector $\boldsymbol{v}_i \in \mathbb{R}^n$ that represents the importance score of the molecule for the $i$th functional group. Combining these vectors constructs an importance matrix $\hat{\boldsymbol{O}} \in \mathbb{R}^{n,P}$, indicating the molecule's relevance across all functional groups. We then compute the loss by comparing the predicted importance matrix with the label matrix, maximizing alignment to enhance focus on key input segments (Equation 11). In this case, the final loss function is shown in Equation 12:

$$L_{algin} = \frac{1}{N} \sum_{i=1}^{N} \frac{1}{nP} ||\boldsymbol{O} - \hat{\boldsymbol{O}}||_F^2, \tag{11}$$

$$L = L_p + \alpha L_{ph} + \beta L_{align}, \tag{12}$$

where $\alpha$ and $\beta$ are controllable parameters.

## 5 EXPERIMENTS

### 5.1 DATASETS

To comprehensively compare SOTA methods, we have curated two benchmark datasets. The first benchmark is Li's molecular property prediction dataset (Li et al., 2023a), consisting eight classification tasks and three regression tasks. The second benchmark, MoleculeACE dataset (van Tilborg et al., 2022), contains thirty regression bio-activity datasets involving activity cliffs.

Additionally, to assess the performance of our models in predicting drug-target interactions, we utilized two distinct datasets from (Song et al., 2023). The first is the BindingDB classification dataset, which focuses on identifying interacting and non-interacting drug-target pairs. The second is the BindingDB regression dataset, which measures interactions' affinity quantitatively. Finally, in our investigation of potential ligand candidates, we have compiled three specific ligand datasets, namely HPK1, FGFR1, and VIM-1. The HPK1 and FGFR1 datasets are from (Li et al., 2023a), while the VIM-1 dataset is sourced from BindingDB, a comprehensive drug database.

### 5.2 EXPERIMENTS CONFIGURATION

In the following sections, we first evaluate the ability of our model, PharmaVQA, to predict molecular properties accurately by comparing its performance with seven methods on widely used molecular property prediction datasets (see Section. 5.3). The datasets were scaffold-split following (Li et al., 2023a) for robust comparison of PharmaVQA's effectiveness in molecule representation. AUC is used for Li's classification dataset, RMSE for Li's regression dataset, and both RMSE and $R^2$ for MoleculeACE datasets.

Furthermore, we evaluate PharmaVQA's ability to discern relationships between molecules and proteins (refer to Section 5.4). In this study, PharmaVQA's performance was tested using the BindingDB classification and regression datasets, and the results were compared to those reported in (Song et al., 2023). For the BindingDB classification dataset, AUC and AUPR were used as metrics, while MSE and Pearson correlation were employed for regression.

To further assess the representation capabilities of PharmaVQA, we applied it to discovering potential ligands for three targets: HPK1, FGFR1, and VIM-1 (see Section 5.5). This experiment focused on identifying ligands for three targets, HPK1, FGFR1, and VIM-1. Performance was measured using Pearson and Spearman correlation coefficients, consistent with benchmarks established in (Li et al., 2023a). Additionally, we evaluated PharmaVQA's ability to associate pharmacophores with the corresponding textual information, demonstrating its capacity to provide valuable insights and enhance interpretability (see Section. 5.6). Comprehensive experiments highlight various aspects of PharmaVQA's remarkable performance, demonstrating its effectiveness across diverse molecular representation tasks.

We address pharmacophore-related questions by shifting the focus from binary classification to regression. Instead of asking whether pharmacophores exist, which could bias results positively, we ask how many are present. Thus, we identified seven common pharmacophores via RDKit and designed specific question templates for each. To enrich semantics, descriptive attributes were added to the questions. For certain pharmacophores, we formulated two distinct questions to capture their unique characteristics and applications, as summarized in Table 5.

Table 1: Performance of property prediction on Li's eight classification datasets using AUC. Results are reported as mean (standard deviation) over three runs with different seeds using scaffold split. Top-1 results are highlighted in bold, and the second best are underlined.

| Methods | BACE | BBBP | ClinTox | SIDER | Estrogen | MetStab | Tox21 | ToxCast |
|---|---|---|---|---|---|---|---|---|
| GraphLoG (Xu et al., 2021) | 0.830(0.014) | 0.846(0.008) | 0.667(0.021) | 0.615(0.013) | 0.871(0.054) | 0.850(0.080) | 0.796(0.025) | 0.677(0.019) |
| GROVER (Rong et al., 2020) | 0.840(0.030) | 0.887(0.006) | 0.874(0.048) | 0.638(0.005) | 0.892(0.044) | 0.876(0.038) | 0.838(0.017) | 0.696(0.014) |
| MolCLR (Wang et al., 2022) | 0.796(0.057) | 0.914(0.015) | 0.869(0.048) | 0.615(0.018) | 0.808(0.085) | 0.814(0.110) | 0.773(0.038) | 0.622(0.010) |
| MoleculeSTM (Liu et al., 2023) | 0.812(0.008) | 0.880(0.013) | 0.875(0.031) | 0.615(0.018) | 0.876(0.073) | 0.860(0.066) | 0.813(0.023) | 0.730(0.013) |
| MoleBERT (Xia et al., 2023) | 0.843(0.031) | 0.851(0.022) | 0.797(0.074) | 0.615(0.010) | 0.887(0.046) | 0.868(0.051) | 0.832(0.021) | 0.720(0.009) |
| KPGT (Li et al., 2023a) | 0.855(0.011) | 0.908(0.010) | **0.946(0.022)** | 0.649(0.009) | 0.905(0.028) | 0.889(0.047) | 0.848(0.013) | **0.746(0.002)** |
| SPMM (Chang & Ye, 2024) | 0.834(0.016) | 0.914(0.015) | 0.897(0.014) | 0.620(0.010) | 0.905(0.046) | 0.841(0.075) | 0.821(0.020) | 0.708(0.011) |
| PharmaVQA (ours) | **0.876(0.017)** | **0.922(0.013)** | **0.946(0.011)** | **0.655(0.023)** | **0.913(0.045)** | **0.892(0.047)** | **0.850(0.029)** | 0.735(0.002) |

Therefore, to explore the advantages of incorporating pharmacophore information, we conducted two ablation studies: First, we evaluated whether incorporating pharmacophores into the VQA query improves performance by comparing a baseline model without pharmacophore questions to one that includes them, using the phrase "to be or not to be, it's a question" as a noise prompt. Second, we examined the impact of querying with multiple pharmacophores compared to querying with one pharmacophores by evaluating both scenarios using our model PharmaVQA. To evaluate performance, we conducted tests across seven pharmacophore types using the molecular property prediction datasets from Li et al. (2023a), as detailed in Table 12 and Table 13. For classification datasets, performance is measured using the Area Under the Curve (AUC), while regression datasets are evaluated using the Root Mean Squared Error (RMSE).

Experimental parameter settings and ablation study results are in Appendix B, G.

## 5.3 EVALUATION OF MOLECULE PROPERTY PREDICTION

In this section, we have conducted a comparative analysis between Li's datasets and the MoleculeACE datasets. The results of these comparisons are summarized in Table 1, Table 2, and Appendix C, providing a comprehensive view of the model's performance across different evaluation frameworks. Each dataset was analyzed three times with distinct random seeds. The baseline models encompass two categories, including those that specialize in molecular graph pre-training such as GraphLoG (Xu et al., 2021), GROVER (Rong et al., 2020), MolCLR (Wang et al., 2022), MoleBERT (Xia et al., 2023), and KPGT (Li et al., 2023a). Additionally, we also consider multimodal models like MoleculeSTM (Liu et al., 2023) and SPMM (Chang & Ye, 2024), which integrate multiple modalities for their unique capabilities. This diverse set of models allows us to comprehensively evaluate our model's performance in various aspects of molecular representation learning. Considering that previous research utilized diverse evaluation settings, we replicated all models under KPGT experimental conditions, excluding KPGT itself and the two models GraphLoG and GROVER, as detailed in the KPGT study.

As shown in Table 1 and Table 2, our model has demonstrated remarkable performance across a diverse range of benchmarks, outperforming several SOTA approaches on both classification and regression tasks. Specifically, when evaluated on Li's eight classification datasets, our model achieved superior AUC scores on seven datasets and ranked 2 on one datasets, consistently ranking among the top performers. Furthermore, when tested on Li's three regression datasets, our model also shone brightly. As demonstrated in the results presented in Appendix C, which encompasses the evaluation of thirty regression datasets from MoleculeACE, the pharmaVQA has exhibited remarkable versatility and robustness across a wide spectrum of molecular regression tasks. Notably, we have achieved outstanding performance, with the lowest RMSE in 24 out of 30 datasets and the highest correlation coefficients ($R^2$) in 23 out of 30 datasets, further underscoring the effectiveness of our method.

## 5.4 EVALUATION OF DRUG-TARGET INTERACTION AND AFFINITY PREDICTION

We conducted experiments on both BindingDB classification and regression datasets. The results of the baseline models are from (Song et al., 2023). Firstly, on the BindingDB classification dataset (as shown in Table 3), PharmaVQA exhibits the most outstanding performance, reaching a top level of

Table 2: Performance of property prediction on Li's three regression datasets using RMSE. Results are reported as mean (standard deviation) over three runs with different seeds using scaffold split. Top-1 results are highlighted in bold, and the second best are underlined.

| Methods | Lipo | Esol | Freesolv |
|---|---|---|---|
| GraphLoG (Xu et al., 2021) | 1.104(0.024) | 2.335(0.073) | 4.174(1.077) |
| GROVER (Rong et al., 2020) | 0.752(0.010) | 0.928(0.027) | 2.991(1.052) |
| MolCLR (Wang et al., 2022) | 0.729(0.052) | 1.249(0.082) | 2.741(0.408) |
| MoleculeSTM (Liu et al., 2023) | 0.706(0.032) | 1.161(0.078) | 3.244(0.634) |
| MoleBERT (Xia et al., 2023) | 0.690(0.023) | 1.185(0.083) | 2.801(0.602) |
| KPGT (Li et al., 2023a) | 0.600(0.010) | **0.803(0.008)** | 2.121(0.837) |
| SPMM (Chang & Ye, 2024) | 0.690(0.029) | 0.872(0.054) | 2.131(0.790) |
| PharmaVQA (ours) | **0.590(0.016)** | 0.841(0.026) | **1.921(0.859)** |

Table 3: Classification performance of PharmaVQA versus six methods on the BindingDB dataset.

| DTI | AUC | AUPR |
|---|---|---|
| GraphDTA (Nguyen et al., 2020) | 0.929 | 0.917 |
| DrugVQA (Zheng et al., 2020) | 0.936 | 0.928 |
| TransformerCPI (Chen et al., 2020) | 0.951 | 0.949 |
| CoaDTI (Huang et al., 2022) | 0.959 | 0.957 |
| MINN-DTI (Li et al., 2022a) | 0.961 | 0.970 |
| PMF-CPI (Song et al., 2023) | 0.990 | 0.990 |
| PharmaVQA (ours) | **0.991** | **0.991** |

Table 4: Regression performance of PharmaVQA versus four methods on the BindingDB dataset.

| DTA | MSE | Pearson |
|---|---|---|
| DeepAffinity (Karimi et al., 2019) | 0.548 | 0.840 |
| DeepDTA (Öztürk et al., 2018) | 0.612 | 0.848 |
| MONN (Li et al., 2020) | 0.584 | 0.858 |
| PMF-CPI (Song et al., 2023) | 0.474 | 0.884 |
| PharmaVQA (ours) | **0.453** | **0.890** |

0.991 in both AUC and AUPR, demonstrating outstanding performance in BindingDB classification prediction tasks. Furthermore, for the comparison on the BindingDB regression dataset (as shown in Table 4), although all methods have displayed some predictive ability, PharmaVQA once again stands out with the lowest MSE value of 0.453 and the highest Pearson value of 0.890, significantly better than other methods. This result indicates that our method has higher accuracy and stronger correlation in predicting the affinity between drugs and targets, providing a more reliable basis for drug design and discovery. In summary, our method demonstrates excellent performance in both BindingDB classification and regression tasks, proving its effectiveness in drug development.

## 5.5 DISCOVERY OF POTENTIAL LIGANDS

To validate that our representation model can effectively uncover potential ligands, we conducted a series of experiments as follows.

**Predicting Binding Affinity on three Ligand Datasets.** Initially, we trained our model on binding affinity datasets of three targets, HPK1, FGFR1, and VIM-1. Hematopoietic progenitor kinase 1 (HPK1) plays a pivotal role in negatively regulating immune functions (Si et al., 2020). Fibroblast growth factor receptor (FGFR1) is a transmembrane receptor tyrosine kinase that is frequently over-expressed or mutated in various diseases such as myeloproliferative syndromes and multiple cancers (Acevedo et al., 2007; Nguyen et al., 2013). Lastly, Verona integron-encoded metallo-$\beta$-lactamase 1 (VIM-1) can hydrolyze carbapenem $\beta$-lactam antibiotics, which leads to serious drug-resistant infections (Boyd et al., 2020).

The prediction results were subsequently compared with the current leading method to assess performance, as detailed in Appendix D. Our model distinctly surpasses the KPGT method across all three ligand datasets with respect to both Spearman and Pearson correlation coefficients. Specifically, HPK1 demonstrated enhancements of 0.032 and 0.024 in the Spearman and Pearson correlation coefficients, respectively. FGFR1 showed improvements of 0.035 and 0.018 for these metrics. Lastly, VIM-1 experienced increases of 0.017 in Spearman and 0.010 in Pearson correlation coefficients.

**Finding Potential Ligands on FDA-approved Dataset.** To highlight the effectiveness of our model in identifying potential ligands, we utilized the trained model to uncover potential ligands among FDA-approved compounds from DrugBank. Specifically, we used the model to predict binding affinity and rank the results. We analyzed the Top-20 molecules with the highest ligand potential

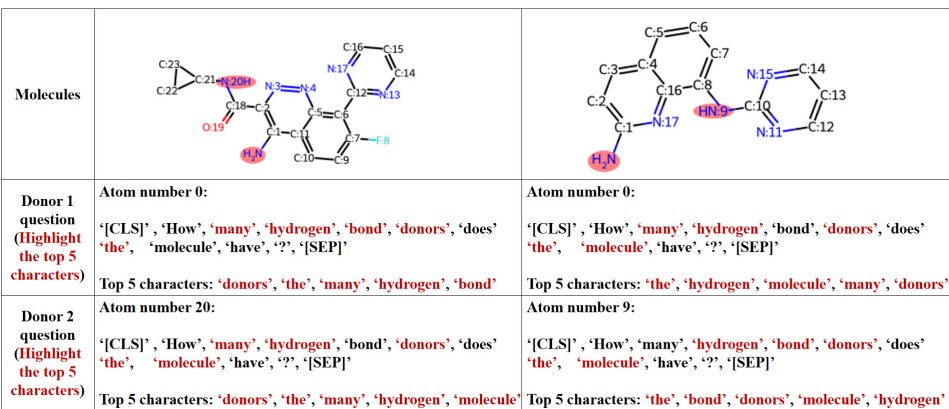

Figure 2: Ligands identified by PharmaVQA with the lowest docking scores.

Figure 3: Visualization of molecules for the Donor question. The top row shows molecules with highlighted donor atoms, while the middle and bottom rows display sorted query characters related to donor atoms, highlighting the Top-5.

by searching for supporting evidence for their roles. Our initial exploration involved a comparative analysis of two significant targets, HPK1 and FGFR1, which are interested in the KPGT model. Interestingly, our model identified 10 and 15 potential ligands, respectively, among the Top-20 predicted molecules, whereas KPGT, as published, identified 12 and 13. This initial analysis demonstrates our approach's competitive advantage.

Besides, we explored molecules from DrugBank targeting VIM-1, an important zinc ion-binding protein. VIM-1 poses a distinct challenge due to its interaction with zinc ions. Our model identified 16 of the Top-20 molecules interacting with zinc ion-binding proteins, proving its reliability and precision. Details of these Top-20 molecules are in Appendix E. Moreover, our model identified six HPK1 ligands and four unique FGFR1 ligands not found by KPGT, demonstrating its distinct ability to explore and identify potential ligands. We also visualized the ligands with the lowest docking scores for each of the three targets (PDB IDs: 5A4C for FGFR1, 7SIU for HPK1, and 5N5H for VIM-1), shown in Figure 2. Additional visualizations of unique ligands are in Appendix F.

## 5.6 CASE STUDY FOR MODEL INTERPRETABILITY

We analyzed the attention map within the final layer of the BAN module to ascertain its proficiency in highlighting vital information related to pharmacophore questions. For this investigation, if the model assigns significant weight to relevant textual information when questioning pharmacophores, it would serve as an indication that the model is adept at extracting pharmacophore-related textual cues, thereby demonstrating its ability to mine meaningful information about drug efficacy.

In this study, we focused on training a model using the Lipo training dataset and then derived pharmacophore-related attention maps for the Lipo test set. Our model produced a donor-specific

attention map for the task of identifying pharmacophores related to donors. This attention map yielded a weight vector relevant to text tokens corresponding to nodes. By sorting this vector and highlighting the Top-5 tokens of related pharmacophore-related atoms, we sought to determine if essential textual information pertinent to donor pharmacophores was among these high-ranking tokens. As shown in Figure 3, it clearly demonstrates that our model effectively captures essential textual information related to donor atoms. Specifically, it significantly emphasizes important terms such as 'hydrogen', 'bond', and 'donors', suggesting that the model has adeptly recognized these as critical components within the framework of donor pharmacophores.

## 6    CONCLUSIONS

In this work, we propose PharmaVQA, a novel deep-learning framework designed to enhance drug discovery by integrating information retrieval techniques to extract pharmacophore-related molecule features. PharmaVQA simplifies feature extraction by directly retrieving key data from molecular libraries, thereby providing more precise insights into drug-target interactions. By employing VQA to construct knowledge prompts, our model enriches molecular representations, ultimately improving the quality of data utilized in downstream tasks. Experimental results across 46 datasets demonstrate PharmaVQA's superiority over existing methods, highlighting its strong generalization capabilities and effectiveness in diverse settings. PharmaVQA's practical utility in drug discovery was validated through experiments on three ligand datasets (HPK1, FGFR1, and VIM-1). Notably, a significant proportion of molecules of PharmaVQA's Top-20 predictions derived from the FDA-approved molecule dataset, have been verified as viable ligands confirmed by literature reports, indicating its potential for identifying promising drug candidates. Case study experiments reveal PharmaVQA's ability to identify meaningful characters in related questions corresponding to pharmacophores, thus showcasing the model's interpretability. Future work will focus on expanding PharmaVQA to include additional molecular interaction factors, as well as incorporating 3D structural data to further enhance prediction accuracy (Li et al., 2024). Additionally, integrating more diverse datasets could broaden its applicability to drug design, enabling more comprehensive analyses. Overall, PharmaVQA presents a promising advancement in drug discovery and design, combining innovative information retrieval with practical, real-world applications.

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

## A PHARMACOPHORES-BASED QUESTION DESIGN

In this section, we formulate questions related to pharmacophores. Due to the ubiquitous presence of pharmacophores in most molecules, constructing samples based on the question of whether pharmacophores exist would result in a significant positive sample bias. To avoid this issue, we reformulated the question to inquire about the number of pharmacophores present, thereby transforming the binary classification problem into a regression problem. This change enabled the model to learn from informative examples. We then used the RDkit tool to identify seven common pharmacophores within the molecules and created specific question templates for each. To enhance semantic richness, we incorporated descriptive textual attributes related to the pharmacophores in the questions, extending beyond simple enumeration. Additionally, to diversify the question texts, we formulated two distinct questions for some pharmacophores, aiming to explore their unique characteristics and potential applications from various angles. Table 5 presents a detailed overview of questions tailored to different pharmacophore designs.

Table 5: Questions on seven different pharmacophores.

| Pharmacophores | Questions |
| --- | --- |
| Donor | How many strongly electronegative atoms that is covalently bonded to hydrogen atoms does the molecule have? |
| | How many hydrogen bond donors does the molecule have? |
| Acceptor | How many electronegative atoms that has at least one available lone pair does the molecule have? |
| | How many hydrogen bond Acceptors does the molecule have? |
| NegIonizable | How many atoms with negatively charges does this molecule have? |
| | How many negative ionized groups does the molecule have? |
| PosIonizable | How many positive ionized groups does the molecule have? |
| Aromatic | How many Aromatic rings does the molecule have? |
| Hydrophobe | How many continuous lipophilic contribution atoms that are not connected to charged atoms does the molecule include? |
| LumpedHydrophobe | How many continuous lipophilic contribution atoms that are not connected to charged atoms or electronegative center inring does the molecule include? |

## B IMPLEMENTATION DETAILS

All experiments conducted in this study are executed utilizing the PyTorch deep learning framework, leveraging a single GPU which is NVIDIA GeForce RTX 4090. The training process is designed with $50$ epochs, with an early stopping criterion of $20$ epochs to prevent overfitting. The graph encoder employed in this work is a pre-trained encoder from (Li et al., 2023a), maintaining the same parameters as outlined in the original paper. For the text encoder, the pre-trained SciBERT is implemented from (Liu et al., 2023), following its original parameter configurations.

To save memory usage, our strategy involves freezing the main part of the $Encoder_g(\cdot)$ used for molecular representation extraction and the $Encoder_t(\cdot)$ used for textual representation extraction. This allows only the top Multi-Layer Perceptron (MLP) layers to be trained. This method preserves the foundational information embedded in these encoders while fine-tuning selectively layers most relevant to the task. Besides, the entire encoder $Encoder_g'(\cdot)$ is trained to fully adapt to downstream tasks.

The batch size is set to $8$. The projection, answering and fusion modules are all Multi-Layer Perceptron (MLP) consisting of $2$ layers, employing the GELU activation function. In the bilinear attention module, we set the parameter of a glimpse to $4$, and the hidden layer dimension was $768$. The parameters $\alpha$ and $\beta$ are consistently set to $1$ across Li's molecular property prediction datasets, BindingDB classification and regression datasets, and three ligand datasets. For the MoleculeACE benchmark datasets, $\alpha$ and $\beta$ are adjusted to $0.5$ and $0.1$, respectively.

## C COMPARISON PHARMAVQA WITH OTHER METHODS ON MOLECULEACE BENCHMARK

Table 6 and Table 7 display the results of our method against other methods, examined using two metrics: RMSE and $R^2$ score over thirty datasets in the MoleculeACE benchmark. These tables highlight our approach's superior performance in predicting molecules' $K_i$ or EC50 values.

Table 6: RMSE results of PharmaVQA compared with seven models on thirty MolecularACE datasets.

| RMSE | GraphLoG | MolCLR | GROVER | MoleculeSTM | MolBERT | KPGT | SPMM | PharmaVQA (ours) |
|---|---|---|---|---|---|---|---|---|
| CHEMBL204Ki | 0.923(0.005) | 0.898(0.003) | 1.017(0.125) | 0.857(0.007) | 0.956(0.010) | 0.678(0.013) | 0.818(0.006) | **0.667(0.006)** |
| CHEMBL214Ki | 0.754(0.015) | 0.716(0.005) | 0.953(0.091) | 0.703(0.004) | 0.753(0.014) | **0.633(0.003)** | 0.765(0.005) | 0.641(0.004) |
| CHEMBL218EC50 | 0.769(0.020) | 0.733(0.004) | 0.793(0.028) | 0.755(0.029) | 0.771(0.011) | 0.654(0.002) | 0.787(0.015) | **0.639(0.001)** |
| CHEMBL219Ki | 0.859(0.011) | 0.843(0.003) | 0.941(0.054) | 0.783(0.004) | 0.872(0.011) | 0.700(0.013) | 0.783(0.003) | **0.695(0.006)** |
| CHEMBL228Ki | 0.796(0.012) | 0.804(0.022) | 0.887(0.040) | 0.765(0.013) | 0.833(0.008) | **0.687(0.006)** | 0.720(0.008) | **0.687(0.001)** |
| CHEMBL231Ki | 0.820(0.006) | 0.757(0.021) | 0.789(0.031) | 0.714(0.007) | 0.785(0.019) | 0.646(0.010) | 0.716(0.014) | **0.634(0.002)** |
| CHEMBL233Ki | 0.884(0.008) | 0.800(0.002) | 0.948(0.081) | 0.804(0.016) | 0.862(0.003) | 0.672(0.001) | 0.800(0.010) | **0.671(0.002)** |
| CHEMBL234Ki | 0.764(0.003) | 0.760(0.038) | 0.924(0.016) | 0.743(0.011) | 0.752(0.005) | 0.601(0.009) | 0.692(0.007) | **0.590(0.004)** |
| CHEMBL235EC50 | 0.745(0.009) | 0.698(0.011) | 0.844(0.078) | 0.730(0.008) | 0.740(0.005) | **0.616(0.003)** | 0.660(0.003) | 0.622(0.003) |
| CHEMBL236Ki | 0.904(0.008) | 0.785(0.007) | 0.936(0.042) | 0.762(0.010) | 0.844(0.013) | 0.640(0.003) | 0.798(0.007) | **0.636(0.003)** |
| CHEMBL237EC50 | 1.000(0.024) | 0.956(0.002) | 1.006(0.035) | 0.871(0.017) | 0.915(0.046) | **0.704(0.021)** | 0.850(0.010) | 0.724(0.002) |
| CHEMBL237Ki | 0.840(0.027) | 0.764(0.011) | 0.958(0.116) | 0.742(0.010) | 0.797(0.010) | 0.679(0.004) | 0.789(0.008) | **0.653(0.004)** |
| CHEMBL238Ki | 0.708(0.008) | 0.681(0.007) | 0.789(0.009) | 0.662(0.011) | 0.706(0.008) | 0.565(0.004) | 0.655(0.016) | **0.556(0.004)** |
| CHEMBL239EC50 | 0.777(0.004) | 0.741(0.011) | 0.821(0.008) | 0.772(0.016) | 0.777(0.007) | 0.645(0.009) | 0.717(0.004) | **0.643(0.001)** |
| CHEMBL244Ki | 0.932(0.010) | 0.845(0.036) | 1.117(0.119) | 0.782(0.005) | 0.892(0.010) | 0.667(0.004) | 0.853(0.017) | **0.666(0.009)** |
| CHEMBL262Ki | 0.848(0.010) | 0.859(0.004) | 0.846(0.007) | 0.780(0.005) | 0.781(0.020) | 0.657(0.010) | 0.736(0.007) | **0.646(0.008)** |
| CHEMBL264Ki | 0.727(0.006) | 0.672(0.009) | 0.745(0.048) | 0.663(0.009) | 0.736(0.011) | **0.551(0.004)** | 0.650(0.004) | 0.556(0.000) |
| CHEMBL287Ki | 0.835(0.003) | 0.784(0.071) | 0.843(0.013) | 0.822(0.022) | 0.816(0.006) | 0.723(0.005) | 0.802(0.015) | **0.700(0.001)** |
| CHEMBL1862Ki | 0.802(0.004) | 0.818(0.043) | 0.833(0.026) | 0.757(0.037) | 0.776(0.022) | 0.637(0.006) | 0.750(0.014) | **0.624(0.003)** |
| CHEMBL1871Ki | 0.707(0.013) | 0.712(0.011) | 0.770(0.005) | 0.743(0.009) | 0.727(0.010) | 0.637(0.005) | 0.721(0.004) | **0.624(0.003)** |
| CHEMBL2034Ki | 0.857(0.017) | 0.775(0.013) | 0.724(0.006) | 0.761(0.023) | 0.800(0.023) | 0.676(0.002) | 0.775(0.010) | **0.666(0.002)** |
| CHEMBL2047EC50 | 0.663(0.005) | 0.620(0.025) | 0.870(0.061) | 0.658(0.013) | 0.643(0.009) | 0.580(0.012) | 0.641(0.024) | **0.583(0.001)** |
| CHEMBL2147Ki | 0.943(0.054) | 0.772(0.007) | 0.820(0.015) | 0.689(0.016) | 0.952(0.062) | 0.583(0.003) | 0.840(0.036) | **0.563(0.002)** |
| CHEMBL2835Ki | 0.480(0.007) | 0.462(0.025) | 0.499(0.035) | 0.471(0.022) | 0.466(0.022) | 0.402(0.008) | 0.426(0.011) | **0.387(0.004)** |
| CHEMBL2971Ki | 0.821(0.007) | 0.773(0.022) | 0.754(0.026) | 0.660(0.008) | 0.777(0.024) | 0.599(0.015) | 0.704(0.019) | **0.589(0.004)** |
| CHEMBL3979EC50 | 0.854(0.013) | 0.771(0.005) | 0.898(0.018) | 0.763(0.010) | 0.799(0.010) | 0.681(0.002) | 0.756(0.013) | **0.662(0.002)** |
| CHEMBL4005Ki | 0.705(0.007) | 0.668(0.024) | 0.717(0.013) | 0.652(0.008) | 0.708(0.010) | 0.567(0.004) | 0.670(0.015) | **0.560(0.007)** |
| CHEMBL4203Ki | 0.970(0.006) | 0.914(0.017) | 0.971(0.042) | 0.905(0.014) | **0.799(0.012)** | 0.864(0.012) | 0.916(0.019) | 0.840(0.008) |
| CHEMBL4616EC50 | 0.747(0.006) | 0.690(0.007) | 0.746(0.064) | 0.667(0.008) | 0.748(0.012) | 0.595(0.012) | 0.716(0.030) | **0.574(0.012)** |
| CHEMBL4792Ki | 0.818(0.014) | 0.788(0.018) | 0.807(0.015) | 0.744(0.015) | 0.800(0.014) | 0.614(0.006) | 0.744(0.007) | **0.604(0.001)** |

## D BINDING AFFINITY OF PHARMAVQA COMPARED WITH KPGT ON THREE LIGAND DATASETS

The comparison of binding affinity performance between PharmaVQA and KPGT is assessed using both the spearman correlation coefficient and the pearson correlation coefficient. Table 8 presents binding affinity results for three ligand datasets.

## E EVIDENCE OF POTENTIAL LIGANDS FOR FGFR1, HPK1, AND VIM-1 IN THE TOP-20 PREDICTIONS

Our method predicted the Top-20 molecules from the FDA-approved DrugBank dataset and identified relevant literature reports respectively. The results are presented in the following Table 9, Table 10 and Table 11.

## F PROTEIN-LIGAND INTERACTION VISUALIZATION

The following Figure 4, Figure 5 and Figure 6 showcase the interactions between the molecules and the related target. The HPK1 and FGFR1 ligands are newly discovered within the predicted Top-20 ligands candidates on FDA approved dataset, as compared to the KPGT's result, which represents the current SOTA method in this field. For the VIM-1 ligands, we chose to visualize five of the newly discovered molecules to highlight our approach's distinct contributions.

Table 7: $R^2$ results of PharmaVQA compared with seven models on thirty MolecularACE datasets.

| $R^2$ | GraphLoG | MolCLR | GROVER | MoleculeSTM | MolBERT | KPGT | SPMM | PharmaVQA (ours) |
|---|---|---|---|---|---|---|---|---|
| CHEMBL204Ki | 0.643(0.004) | 0.662(0.003) | 0.560(0.112) | 0.692(0.005) | 0.617(0.008) | 0.810(0.008) | 0.720(0.004) | **0.813(0.003)** |
| CHEMBL214Ki | 0.584(0.016) | 0.588(0.066) | 0.331(0.122) | 0.639(0.004) | 0.586(0.015) | 0.706(0.003) | 0.572(0.006) | **0.700(0.003)** |
| CHEMBL218EC50 | 0.437(0.030) | 0.528(0.066) | 0.401(0.042) | 0.456(0.042) | 0.433(0.016) | 0.595(0.003) | 0.409(0.022) | **0.611(0.001)** |
| CHEMBL219Ki | 0.410(0.015) | 0.431(0.004) | 0.290(0.082) | 0.510(0.004) | 0.392(0.016) | 0.612(0.014) | 0.510(0.004) | **0.614(0.006)** |
| CHEMBL228Ki | 0.570(0.013) | 0.561(0.024) | 0.466(0.049) | 0.603(0.013) | 0.529(0.009) | 0.678(0.006) | 0.648(0.008) | **0.680(0.001)** |
| CHEMBL231Ki | 0.590(0.006) | 0.651(0.019) | 0.620(0.030) | 0.689(0.006) | 0.624(0.018) | 0.740(0.008) | 0.688(0.012) | **0.755(0.002)** |
| CHEMBL233Ki | 0.546(0.009) | 0.628(0.002) | 0.474(0.092) | 0.624(0.015) | 0.568(0.003) | 0.735(0.001) | 0.628(0.009) | **0.738(0.001)** |
| CHEMBL234Ki | 0.578(0.004) | 0.654(0.070) | 0.383(0.022) | 0.602(0.011) | 0.591(0.005) | 0.742(0.007) | 0.654(0.007) | **0.749(0.003)** |
| CHEMBL235EC50 | 0.505(0.012) | 0.566(0.013) | 0.361(0.114) | 0.525(0.010) | 0.512(0.006) | **0.657(0.003)** | 0.612(0.004) | 0.655(0.003) |
| CHEMBL236Ki | 0.542(0.008) | 0.655(0.006) | 0.507(0.045) | 0.674(0.008) | 0.600(0.012) | 0.767(0.003) | 0.643(0.006) | **0.773(0.002)** |
| CHEMBL237EC50 | 0.495(0.024) | 0.539(0.002) | 0.488(0.036) | 0.617(0.015) | 0.576(0.043) | **0.739(0.015)** | 0.635(0.009) | 0.735(0.002) |
| CHEMBL237Ki | 0.611(0.025) | 0.679(0.009) | 0.488(0.128) | 0.697(0.008) | 0.650(0.009) | **0.765(0.003)** | 0.658(0.007) | **0.765(0.003)** |
| CHEMBL238Ki | 0.609(0.009) | 0.639(0.008) | 0.515(0.011) | 0.658(0.011) | 0.611(0.009) | 0.751(0.003) | 0.665(0.016) | **0.760(0.004)** |
| CHEMBL239EC50 | 0.517(0.005) | 0.561(0.013) | 0.461(0.011) | 0.523(0.019) | 0.518(0.008) | **0.671(0.010)** | 0.589(0.004) | 0.670(0.001) |
| CHEMBL244Ki | 0.684(0.007) | 0.739(0.022) | 0.542(0.095) | 0.777(0.003) | 0.710(0.007) | 0.836(0.002) | 0.735(0.011) | **0.838(0.005)** |
| CHEMBL262Ki | 0.377(0.014) | 0.361(0.006) | 0.380(0.010) | 0.473(0.006) | 0.471(0.026) | 0.630(0.012) | 0.530(0.009) | **0.638(0.008)** |
| CHEMBL264Ki | 0.538(0.008) | 0.605(0.010) | 0.512(0.065) | 0.616(0.01) | 0.526(0.015) | **0.737(0.004)** | 0.630(0.004) | 0.729(0.001) |
| CHEMBL287Ki | 0.448(0.004) | 0.456(0.009) | 0.437(0.017) | 0.465(0.028) | 0.473(0.007) | 0.585(0.005) | 0.490(0.018) | **0.612(0.001)** |
| CHEMBL1862Ki | 0.684(0.003) | 0.670(0.035) | 0.658(0.021) | 0.718(0.027) | 0.703(0.017) | 0.802(0.004) | 0.723(0.010) | **0.808(0.002)** |
| CHEMBL1871Ki | 0.516(0.018) | 0.508(0.015) | 0.425(0.007) | 0.464(0.013) | 0.488(0.014) | 0.61(0.007) | 0.496(0.005) | **0.623(0.003)** |
| CHEMBL2034Ki | 0.325(0.026) | 0.448(0.018) | 0.456(0.009) | 0.467(0.032) | 0.412(0.033) | 0.588(0.002) | 0.448(0.014) | **0.592(0.003)** |
| CHEMBL2047EC50 | 0.543(0.007) | 0.601(0.032) | **0.782(0.031)** | 0.550(0.018) | 0.571(0.012) | 0.640(0.014) | 0.572(0.032) | 0.647(0.001) |
| CHEMBL2147Ki | 0.743(0.030) | 0.829(0.003) | 0.807(0.007) | 0.863(0.006) | 0.739(0.033) | 0.902(0.001) | 0.797(0.017) | **0.909(0.001)** |
| CHEMBL2835Ki | 0.744(0.007) | 0.763(0.026) | 0.722(0.039) | 0.753(0.022) | 0.759(0.022) | 0.822(0.007) | 0.798(0.011) | **0.833(0.004)** |
| CHEMBL2971Ki | 0.661(0.006) | 0.699(0.017) | 0.714(0.020) | 0.781(0.006) | 0.696(0.019) | 0.824(0.009) | 0.751(0.013) | **0.826(0.003)** |
| CHEMBL3979EC50 | 0.413(0.018) | 0.523(0.006) | 0.352(0.026) | 0.532(0.013) | 0.487(0.013) | 0.632(0.002) | 0.540(0.016) | **0.648(0.002)** |
| CHEMBL4005Ki | 0.504(0.009) | 0.554(0.031) | 0.488(0.018) | 0.575(0.010) | 0.500(0.014) | 0.677(0.004) | 0.552(0.020) | **0.687(0.008)** |
| CHEMBL4203Ki | 0.182(0.009) | 0.273(0.026) | 0.178(0.072) | 0.288(0.022) | **0.444(0.017)** | 0.338(0.017) | 0.269(0.03) | 0.386(0.012) |
| CHEMBL4616EC50 | 0.357(0.010) | 0.452(0.011) | 0.355(0.112) | 0.488(0.012) | 0.357(0.020) | 0.585(0.016) | 0.409(0.050) | **0.621(0.015)** |
| CHEMBL4792Ki | 0.410(0.020) | 0.452(0.025) | 0.426(0.022) | 0.512(0.019) | 0.435(0.020) | 0.664(0.007) | 0.512(0.009) | **0.678(0.000)** |

Table 8: Binding affinity results of PharmaVQA compared with KPGT on three ligand datasets. The Spearman correlation coefficient (Spearman) and Pearson correlation coefficient (Pearson) are used as metrics.

| Methods | HPK1 | | FGFR1 | | VIM1 | |
|---|---|---|---|---|---|---|
| | Spearman | Pearson | Spearman | Pearson | Spearman | Pearson |
| KPGT | 0.866(0.009) | 0.908(0.004) | 0.901(0.002) | 0.924(0.001) | 0.886(0.047) | 0.931(0.015) |
| PharmaVQA (ours) | **0.898(0.013)** | **0.932(0.003)** | **0.936(0.022)** | **0.942(0.017)** | **0.903(0.035)** | **0.941(0.015)** |

Table 9: Top-20 predicted potential HPK1 ligands with source by PharmaVQA.

| No. | Drugbank | Name | Source |
|---|---|---|---|
| 1 | DB06616 | Bosutinib | $K_d$ = 15 nM (Davis et al., 2011) |
| 2 | DB12267 | Brigatinib | Not found |
| 3 | DB01268 | Sunitinib | $K_i$ = 16 nM (Davis et al., 2011) |
| 4 | DB12332 | Rucaparib | Not found |
| 5 | DB09073 | Palbociclib | Not found |
| 6 | DB12500 | Fedratinib | $K_i$ = 9 nM (U.S.Patent 2018183964A1) |
| 7 | DB09063 | Ceritinib | $K_i$ = $2e^{-5}$ uM (U.S.Patent WO2018183956A1) |
| 8 | DB11828 | Neratinib | $K_d$ = 16 nM (Davis et al., 2011) |
| 9 | DB09079 | Nintedanib | $IC_{50}$ = 45 nM (U.S.Patent WO2019238067A1) |
| 10 | DB15685 | Selpercatinib | Not found |
| 11 | DB08881 | Vemurafenib | $K_i$ = $1.13e^{-4}$ uM (U.S.Patent WO2018183956A1) |
| 12 | DB09330 | Osimertinib | Not found |
| 13 | DB12010 | Fostamatinib | $K_d$ = 72 nM (Karaman et al., 2008) |
| 14 | DB06595 | Midostaurin | $K_d$ = 2100 nM (Davis et al., 2011) |
| 15 | DB12141 | Gilteritinib | Not found |
| 16 | DB11963 | Dacomitinib | $K_i$ = $4.76e^{-4}$ uM (U.S.PatentWO2018183956A1) |
| 17 | DB12887 | Tazemetostat | Not found |
| 18 | DB08912 | Dabrafenib | Not found |
| 19 | DB00762 | Irinotecan | Not found |
| 20 | DB09027 | Ledipasvir | Not found |

Table 10: Top-20 predicted potential FGFR1 ligands with source by PharmaVQA.

| No. | Drugbank | Name | Source |
|---|---|---|---|
| 1 | DB12147 | Erdafitinib | $IC_{50}$ = 1.2 nM (Perera et al., 2017) |
| 2 | DB15149 | Futibatinib | (Sootome et al., 2020) |
| 3 | DB08881 | Vemurafenib | (Zhang et al., 2016) |
| 4 | DB08901 | Ponatinib | $IC_{50}$ = 0.7 nM (Gao et al., 2018) |
| 5 | DB08875 | Cabozantinib | $IC_{50}$ = 11.3 nM (Li et al., 2021) |
| 6 | DB09079 | Nintedanib | $IC_{50}$ = 45 nM (U.S. PatentWO2019238067A1) |
| 7 | DB11886 | Infigratinib | $IC_{50}$ = 1.1 nM (Kang, 2021) |
| 8 | DB15102 | Pemigatinib | $IC_{50}$ = 0.4 nM (Wu et al., 2021) |
| 9 | DB12267 | Brigatinib | $IC_{50}$ = 0.109 $\mu$M by Cell titer Glo assay (Yang et al., 2023) |
| 10 | DB09078 | Lenvatinib | $IC_{50}$ = 46 nM (Matsui et al., 2008) |
| 11 | DB06589 | Pazopanib | $IC_{50}$ = 0.15 nM (Karaman et al., 2008) |
| 12 | DB11853 | Relugolix | Not found |
| 13 | DB12010 | Fostamatinib | $K_d$ = 72 nM (Karaman et al., 2008) |
| 14 | DB11986 | Entrectinib | $IC_{50}$ = 1uM (Menichincheri et al., 2016) |
| 15 | DB08865 | Crizotinib | $IC_{50}$ = 1000 nM (Liu et al., 2016) |
| 16 | DB11718 | Encorafenib | Not found |
| 17 | DB11800 | Tivozanib | (Vijayan et al., 2015) |
| 18 | DB14840 | Ripretinib | Not found |
| 19 | DB15444 | Elexacaftor | Not found |
| 20 | DB09042 | Tedizolid phosphate | Not found |

Table 11: Top-20 predicted potential VIM-1 ligands with source by PharmaVQA.

| No. | Drugbank | Name | Zinc ion binding | Source |
|---|---|---|---|---|
| 1 | DB11326 | Boric acid | Yes | (Yusuf et al., 2022) |
| 2 | DB11127 | Selenious acid | No | Not found |
| 3 | DB08906 | Futicasone furoate | Yes | (Issar et al., 2006) |
| 4 | DB00932 | Tipranavir | Yes | (Hudon et al., 2008; Zhou et al., 2023) |
| 5 | DB08881 | Vemurafenib | Yes | (Satow et al., 2022) |
| 6 | DB00588 | Fluticasone propionate | Yes | (Issar et al., 2006) |
| 7 | DB14669 | Betamethasone phosphate | Yes | (Czock et al., 2005) |
| 8 | DB13055 | Oteseconazole | Yes | (Warrilow et al., 2017) |
| 9 | DB13867 | Fluticasone | Yes | (Issar et al., 2006) |
| 10 | DB09041 | Tavaborole | Yes | (Bonardi et al., 2020) |
| 11 | DB14631 | Prednisolone phosphate | Yes | (Czock et al., 2005) |
| 12 | DB09378 | Fluprednisolone | No | Not found |
| 13 | DB00278 | Argatroban | No | Not found |
| 14 | DB14657 | Pharamethasone acetate | No | Not found |
| 15 | DB11921 | Deflazacort | Yes | (Möllmann et al., 1995) |
| 16 | DB00596 | Ulobetasol | Yes | (Mohandas et al., 2009) |
| 17 | DB11121 | Chloroxylenol | Yes | (Pottel et al., 2020) |
| 18 | DB14761 | Remdesivir | Yes | (Agostini et al., 2018; Gordon et al., 2020) |
| 19 | DB00800 | Fenoldopam | Yes | (Martin & Broadley, 1995) |
| 20 | DB14542 | Hydrocortisone phosphate | Yes | (GROSSMAN et al., 2006) |

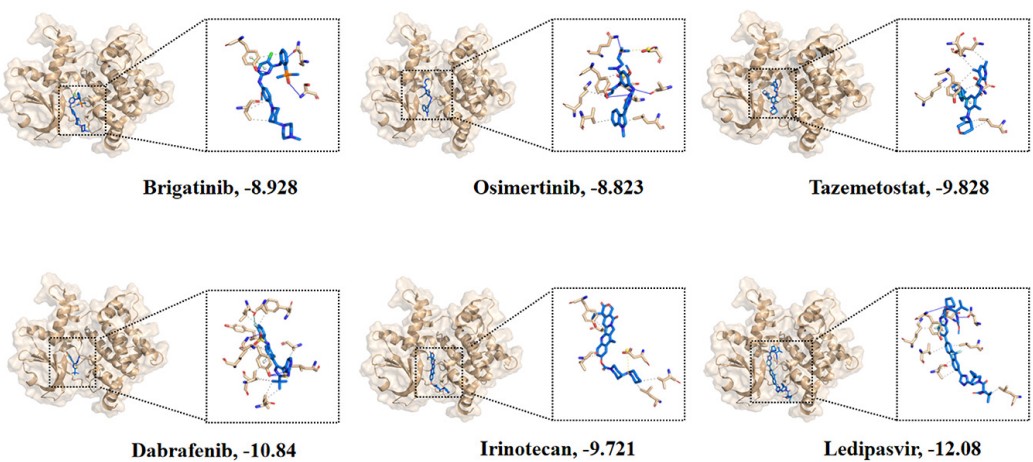

**Brigatinib, -8.928**  **Osimertinib, -8.823**  **Tazemetostat, -9.828**

**Dabrafenib, -10.84**  **Irinotecan, -9.721**  **Ledipasvir, -12.08**

Figure 4: A new set of six potential HPK1 ligands were identified by PharmaVQA (target PDB ID: 7SIU).

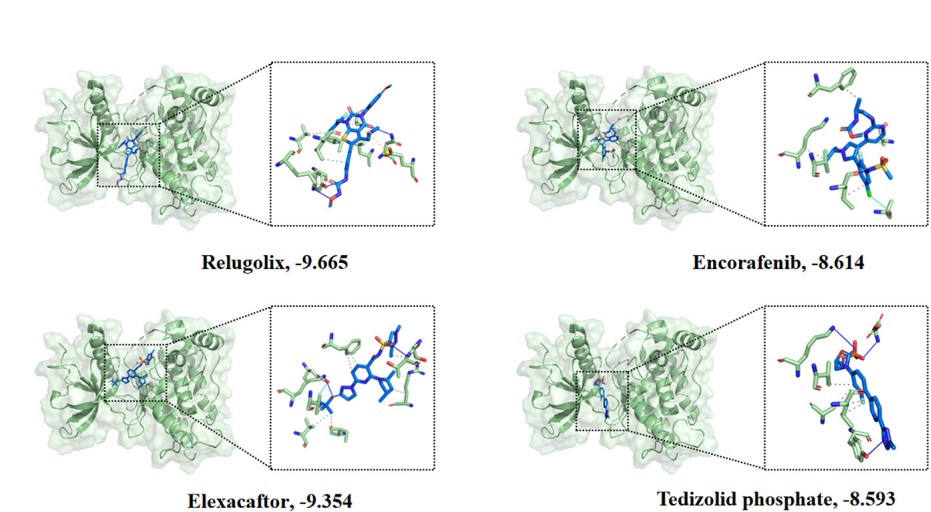

Figure 5: A new set of four potential FGFR1 ligands were identified by PharmaVQA (target PDB ID: 5A4C).

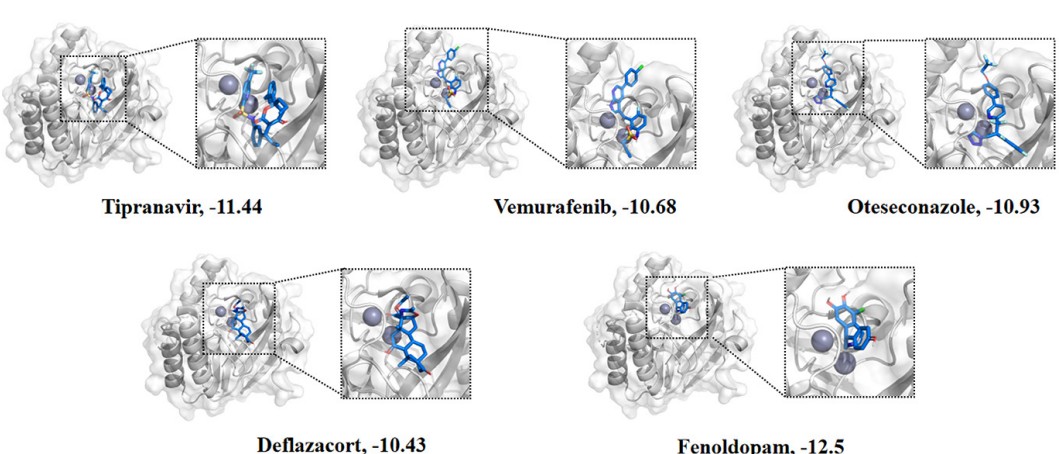

Figure 6: A new set of five potential VIM-1 ligands were identified by PharmaVQA (target PDB ID: 5N5H).

# G    ABLATION STUDY

We conducted two ablation studies to gain deeper insights into the effectiveness of leveraging pharmacophore information into our VQA model as retrieval knowledge. Firstly, we aimed to validate whether leveraging pharmacophores as part of the query in VQA is beneficial. This experiment compared the performance of a baseline VQA model, which does not utilize pharmacophore-related questions, against a modified model specifically designed to incorporate pharmacophore questions. Specifically, we utilize the sentence: "to be or not to be, it's a question." as the noise question.

Secondly, we sought to investigate the difference in performance between querying with multiple pharmacophores simultaneously versus querying with a single pharmacophore individually. This experiment involved running our pharmacophore-integrated VQA model on two scenarios: one containing queries with a single pharmacophore and the other with queries encompassing multiple pharmacophores. Specifically, We use the VQA module separately based on the seven different pharmacophore types, testing these performances on Li's molecular property prediction datasets, which is shown in Table 12 and Table 13. The metric used for Li's classification dataset is the Area Under the Curve (AUC), while for Li's regression dataset, the Root Mean Squared Error (RMSE) is employed.

Table 12: An ablation study was conducted on Li's eight classification datasets by PharmaVQA, evaluating noise queries, single queries, and all seven queries using AUC as the metric.

|  | BACE | BBBP | ClinTox | SIDER | Estrogen | MetStab | Tox21 | ToxCast |
|---|---|---|---|---|---|---|---|---|
| Noise question | 0.846(0.017) | 0.901(0.023) | 0.906(0.029) | 0.649(0.017) | 0.905(0.046) | 0.872(0.046) | 0.842(0.020) | 0.729(0.009) |
| Single question | 0.863(0.015) | 0.912(0.014) | 0.908(0.033) | 0.645(0.022) | 0.910(0.046) | 0.886(0.060) | 0.837(0.023) | 0.731(0.006) |
| All question | **0.876(0.017)** | **0.922(0.013)** | **0.946(0.011)** | **0.655(0.023)** | **0.913(0.045)** | **0.892(0.047)** | **0.850(0.029)** | **0.735(0.002)** |

Table 13: An ablation study was conducted on Li's three regression datasets by PharmaVQA, evaluating noise queries, single queries, and all seven queries using RMSE as the metric.

|  | Lipo | Esol | Freesolv |
|---|---|---|---|
| Noise question | 0.609(0.026) | 0.999(0.039) | 2.202(1.108) |
| Single question | 0.598(0.011) | 0.971(0.068) | 2.030(1.071) |
| All question | **0.590(0.016)** | **0.841(0.026)** | **1.921(0.859)** |

By comparing the performance metrics across these two scenarios, it reveals that utilizing multiple pharmacophore questions concurrently offers superior results than isolating a single pharmacophore question. These findings from the ablation study illuminate the best approach to integrating pharmacophore information into VQA models. Detailed AUC and RMSE results for seven distinct pharmacophore questions are presented in Tables 14 and 15.

Table 14: The AUC Results for each pharmacophore question across Li's eight classification datasets of PharmaVQA.

|  | Donor | Accepter | NegIonizable | PosIonizable | Aromatic | Hydrophobe | LumpedHydrophobe |
|---|---|---|---|---|---|---|---|
| BACE | 0.856(0.027) | 0.863(0.012) | 0.873(0.011) | 0.865(0.004) | 0.852(0.019) | 0.862(0.018) | 0.869(0.012) |
| BBBP | 0.911(0.013) | 0.917(0.018) | 0.911(0.008) | 0.906(0.014) | 0.916(0.017) | 0.914(0.017) | 0.913(0.016) |
| ClinTox | 0.916(0.035) | 0.912(0.026) | 0.904(0.047) | 0.901(0.035) | 0.908(0.012) | 0.885(0.059) | 0.928(0.019) |
| SIDER | 0.645(0.030) | 0.638(0.024) | 0.649(0.019) | 0.647(0.019) | 0.651(0.016) | 0.644(0.026) | 0.643(0.024) |
| Estrogen | 0.911(0.039) | 0.917(0.039) | 0.913(0.043) | 0.907(0.056) | 0.911(0.045) | 0.908(0.046) | 0.902(0.052) |
| MetStab | 0.880(0.061) | 0.879(0.064) | 0.872(0.016) | 0.878(0.022) | 0.879(0.064) | 0.897(0.049) | 0.872(0.070) |
| Tox21 | 0.830(0.020) | 0.836(0.029) | 0.835(0.015) | 0.839(0.016) | 0.836(0.021) | 0.834(0.025) | 0.836(0.025) |
| ToxCast | 0.730(0.005) | 0.734(0.009) | 0.726(0.009) | 0.731(0.009) | 0.729(0.004) | 0.732(0.007) | 0.734(0.002) |

Table 15: The RMSE Results for each pharmacophore question across Li's three regression datasets of PharmaVQA.

|  | Donor | Accepter | NegIonizable | PosIonizable | Aromatic | Hydrophobe | LumpedHydrophobe |
|---|---|---|---|---|---|---|---|
| Lipo | 0.614(0.008) | 0.594(0.025) | 0.602(0.008) | 0.599(0.006) | 0.596(0.010) | 0.595(0.014) | 0.588(0.009) |
| Esol | 0.909(0.087) | 1.027(0.100) | 0.896(0.021) | 0.981(0.115) | 0.993(0.088) | 1.018(0.047) | 0.973(0.022) |
| Freesolv | 2.040(1.053) | 1.985(1.029) | 2.071(1.241) | 2.036(1.027) | 2.000(1.085) | 1.979(1.027) | 2.100(1.032) |

