# OpenReview forum: "PharmaVQA: A Retrieval-Augmented Visual Question Answering Framework for Molecular Representation via Pharmacophore Guided Prompts"
_ICLR.cc/2025/Conference — Submitted to ICLR 2025_

### Official Review · Reviewer_zSST · 2024-10-28

**Soundness:** 3
**Presentation:** 2
**Contribution:** 3
**Rating:** 5
**Confidence:** 4

**Summary:**

The paper introduces a framework to extract pharmacophore information from molecules with text question answering. The learned embeddings are then connected with embeddings from another molecular encoder for downstream tasks. This framework achieves good results on several downstream tasks.

**Strengths:**

1. The integration of text-based question-answering to enhance molecular representations is an interesting and original idea. By directly querying pharmacophore-related information, the model captures key molecular features that may be overlooked by conventional molecule-text alignment models like MoleculeSTM.

2. PharmaVQA achieves SOTA results on multiple downstream tasks.

**Weaknesses:**

1. Paper writing can be improved.  The paper’s focus on detailed architectural descriptions detracts from its main contributions. Readers might find greater value in a thorough explanation of how the pharmacophore-related questions were designed and in ablation studies that demonstrate the effectiveness of the QA component. Important insights into the QA framework’s impact on downstream tasks are relegated to the appendix, whereas they should be central to the main text.

2. The provided visualizations show that important textual information related to donor pharmacophores ranks highly among tokens, which is expected in molecule-text multimodal models. However, the paper lacks a deeper analysis of how this textual information and pharmacophore knowledge extraction enhance downstream task performance. For example, it would be insightful to explore whether the prompt embeddings generated by specific pharmacophore related questions contribute more to some specific property prediction tasks.

**Questions:**

1. The term “Retrieval-Augmented” in the title seems to refer to the extraction of knowledge from the molecule itself, rather than retrieving external data. Could the authors clarify this usage? If external data retrieval is not involved, the term might be somewhat misleading.

2. For each downstream task, what specific model is used as the downstream encoder ? It would strengthen the paper to include performance comparisons between models using only this encoder without the QA prompt embeddings and those that include them. Such ablation studies are crucial to demonstrate the distinct contribution of the QA component to the overall performance.

3. The current approach trains the feature extraction module and the downstream tasks simultaneously. Is it feasible to train these components separately? Exploring this possibility could provide insights into the modularity of the framework and its applicability to different tasks without retraining the entire model.

---

> ### Author Response · Authors · 2024-11-21
> **Author Response for Weakness (1/1)**
>
> # Response to Weakness 1:
> Thank you for your valuable suggestions. We understand your concerns, and below are our responses regarding the writing of the paper.
>
> - **Focus on pharmacophore-related questions**: We recognize that the design of the pharmacophore-related questions is a crucial part of our framework. To make this clearer, we plan to place greater emphasis on the design of these questions.
> - **Ablation studies and QA component**: Regarding the VQA framework's impact on downstream tasks, we agree that the ablation studies should be integrated into the main text to provide more clarity on the effectiveness of the QA component.
>
> However, we also need to keep in mind that the main body of the paper has a limited page count, so we will prioritize the most critical elements to keep the paper concise and focused. While some of the detailed architectural descriptions and supplementary analyses (such as the detailed results) may remain in the appendix, the most essential insights regarding the design of the pharmacophore questions and the ablation studies will be moved into the main text for better clarity and impact.
>
> # Response to Weakness 2:
>
> We have considered this aspect in our experiments. Table 12 and Table 13: When we artificially made the pharmacophore-related information irrelevant, the model's performance on prediction tasks decreased, highlighting that the model is indeed leveraging this knowledge in a meaningful way. Specifically, we explored the importance of pharmacophore-related tokens through an ablation study where we introduced Noise question as the input question, effectively disturbing the important token information. As a result, we observed a performance drop, indicating that the pharmacophore-related tokens play a significant role in the model’s ability to predict molecular properties.
>
> Regarding the impact of a given pharmacophore question on specific tasks, we conducted further analysis based on Table 14 and Table 15. Table 14 and Table 15 present the model performance on Li's 8 classification datasets and 3 regression datasets when the input consists of a single pharmacophore question.
>
> For example, in the Lipo dataset, the goal is to predict the molecular membrane permeability and solubility, with the target value being the octanol/water partition coefficient. This coefficient reflects the differences in solubility of a molecule in water and an organic solvent (such as octanol), and is commonly used to describe whether a molecule can permeate biological membranes. For such tasks, hydrophobic features of the molecule often have a significant impact on properties like solubility and membrane permeability.
> **From the data in Table 15, it can be observed that when the model is queried with the LumpedHydrophobe pharmacophore, it achieves the best performance on the Lipo dataset, indicating a strong correlation between the LumpedHydrophobe pharmacophore and the task of the dataset**. The LumpedHydrophobe pharmacophore represents molecular segments with hydrophobic characteristics, typically referring to non-polar groups or regions that repel water molecules. Due to its hydrophobic nature, the LumpedHydrophobe pharmacophore interacts with organic solvents like octanol, directly correlating with the octanol/water partition coefficient target. This result further confirms that, with pharmacophore-related questions, the model is able to more effectively extract task-relevant information, thereby enhancing its performance on downstream tasks.
>
> Through these analyses, we further demonstrate that pharmacophore-related questions can effectively enhance the model's understanding of the task. This further validates the potential of pharmacophore questions in molecular representation.

---

> > ### Author Response · Authors · 2024-11-21
> > **Author Response for Questions (1/1)**
> >
> > # Response to Q1:
> > Thank you for your observation. In our case, the term "Retrieval-Augmented" was intended to highlight the process of extracting relevant knowledge from the molecule itself, rather than retrieving external data sources. Specifically, we use a question-answering (QA) mechanism to extract pharmacophore-related knowledge from the molecule’s internal features, which are then used to augment the molecular representation for downstream tasks.
> >
> > For the distinction and relationship with traditional knowledge retrieval, please refer to General Response.
> >
> > # Response to Q2:
> > Thank you for your question. Following this, we discuss the encoders designed for specific downstream tasks and  the ablation studies:
> > - **Downstream task-specific encoders**: The encoder used in our framework is consistent across different downstream tasks. For molecules encoder, we use the pre-trained encoder from [1], for text encoder, we use the pre-trained SciBERT from [2], which helps maintain robustness and reduces resource overhead.
> >
> > - **The ablation studies without the QA prompt embeddings**：We recognize that ablation studies are crucial for validating the contribution of the QA components to the overall model performance. **In the paper, we have demonstrated the contrast between models enhanced by introducing pharmacophore-related questions and models that do not incorporate the QA prompt information**.
> > Specifically, in  Section 5.2 Evaluation of Molecule Property Prediction, we've already compared our model, named PharmaVQA, which incorporates pharmacophore-related QA prompts, with a baseline model that does not. This baseline, the KPGT model, is based on the same molecular representation model without pharmacophore-related prompts. Our findings demonstrate that including pharmacophore-related QA prompts significantly improves performance across various tasks.
> >
> > This study underscores the value of including pharmacophore knowledge in the form of QA prompts, highlighting their positive impact on the model's ability to predict molecular properties effectively.
> >
> > # Response to Q3:
> > Thank you for your suggestion. Training the feature extraction module and downstream tasks separately is an available idea, but there are several reasons we chose joint training in our current approach.
> >
> > First, both the molecular feature extraction and text models (e.g., Line Graph Transformer and SciBERT) are pre-trained and remain frozen during the training phase. In this setup, only a small number of parameters such as the heads for downstream tasks and VQA prompt embeddings are updated. This approach significantly reduces the computational cost while still enabling the model to learn task-specific features effectively.
> >
> > Moreover, the modular design of our framework allows flexibility, as each component (molecular feature extraction, VQA-based pharmacophore extraction, and downstream tasks) can be independently modified or extended. However, joint training ensures that these components work cohesively by fine-tuning them together for specific downstream tasks, improving overall performance and efficiency.
> >
> > # Reference
> > [1].Li, H., Zhang, R., Min, Y., Ma, D., Zhao, D., & Zeng, J. (2023). A knowledge-guided pre-training framework for improving molecular representation learning. Nature Communications, 14(1), 7568.
> >
> > [2].Liu, S., Nie, W., Wang, C., Lu, J., Qiao, Z., Liu, L., ... & Anandkumar, A. (2023). Multi-modal molecule structure–text model for text-based retrieval and editing. Nature Machine Intelligence, 5(12), 1447-1457.

---

> > > ### Comment · Reviewer_zSST · 2024-11-25
> > >
> > > Thank you for the response. The baseline model for PharmaVQA, KPGT, utilizes the same molecular representation framework but without pharmacophore-related prompts. However, as seen in Tables 1 and 2, the performance improvement with PharmaVQA over KPGT appears marginal and may not be statistically significant. In fact, for certain tasks, the use of prompt embeddings results in worse performance, which raises questions about the robustness of the approach.
> > >
> > > Additionally, as another reviewer noted, PharmaVQA does not demonstrate statistically superior results compared to the simpler method of appending pharmacophore-based features derived from RDKit. This observation further calls into question the overall effectiveness and added value of the proposed method.
> > >
> > > Given these concerns, I will maintain my score at 5 for now

---

> ### Author Response · Authors · 2024-11-28
> **Author Response (1/3)**
>
> Dear Reviewer zSST,
>
> Thank you for your detailed and constructive feedback. We are very sorry for unclearly explaining the overall framework of our PharmaVQA architecture and results. To address these concerns, we have provided a detailed explanation from both theoretical and experimental perspectives, as outlined below:
>
> 1. **Justification of pharmacophore-related prompts**
>
>   Firstly, our model PharmaVQA was designed as a Pharmacophore Guided VQA framework for molecular representation. Current machine learning-based methods often overlook the multifaceted nature of compounds, resulting in inaccurate molecular related downstream tasks. However, the process of constructing a comprehensive molecular representations often relatively complex for the learning of compounds feature, particularly when dealing with numerous aspects of molecular features. Consequently, we aim to incorporate Pharmacophore structure to enhanced the chemical and drug-like plausible information. Therefore, we applied the Pharmacophore feature as the guidance for molecular embedding.
>
> Secondly, to address the concern, we conducted the ablation studies with statistical result comparing the performance of models with and without the guidance of pharmacophore. The results (Table 1) on 3 datasets demonstrate that incorporating pharmacophore information enhances model accuracy. Specifically, **with average RMSE increasing by 11% compared to noise question guidance and  7%  compared to single question guidance**. This highlights the critical role of pharmacophore question guidance in refining predictions and ensuring plausible drug-like feature embedding.
>
> Table 1. An ablation study was conducted on Li’s three regression datasets by PharmaVQA, evaluating noise questions, single question, and all seven queries using AUC as the metric.
> | Methods| Lipo            | Esol         | freesolv     |
> |-----------------|-----------------|--------------|--------------|
> | Noise question  | 0.609(0.026) | 0.999(0.039) | 2.202(1.108) |
> | single question | 0.598(0.011) | 0.971(0.068) | 2.03(1.071)  |
> | all question    | **0.59(0.016)**  | **0.841(0.026)** | **1.921(0.859)** |
>
> 2. **Justification on certain tasks performance**
>
> Thank you for pointing out this concern. We apologize for any confusion caused by the unclear presentation of our results.
>
> PharmaVQA is designed as a general framework, evaluated across a diverse range of tasks and datasets to test its robustness and applicability. In total, we conducted experiments on 46 datasets, with observed performance drops limited to only several datasets and being marginal (e.g., differences in rank or metric values within a few percentage points). Specifically, our study includes:
> - 11 datasets from Li’s benchmark: PharmaVQA achieved rank-1 performance on 9 datasets.
> - 30 datasets from MoleculeACE: PharmaVQA achieved rank-1 performance on 24 datasets.
> - 2 datasets from BindingDB and 3 datasets from ligand affinity benchmarks, where PharmaVQA outperformed baseline models in all cases.
>
> Moreover, PharmaVQA provides biologically meaningful outcomes, as demonstrated in applications like identifying potential ligands:
> - On the **FDA-approved dataset for finding potential ligands**, PharmaVQA identified **10 and 15 ligands for HPK1 and FGFR1** (details in Table 9 and Table 10 in manuscript), respectively, among the Top-20 predicted molecules, while KPGT identified **12 and 13 ligands** for these targets. Moreover, our model identified **6 unique HPK1 ligands and 4 unique FGFR1 ligands not found by KPGT**.  Additionally, PharmaVQA successfully identified **16 out of Top-20 potential ligands for VIM-1**  (details in Table 11 in manuscript).
>
> These results could highlight PharmaVQA’s overall robustness and effectiveness in capturing pharmacophore-specific insights.
>
> （to be continued）

---

> > ### Author Response · Authors · 2024-11-28
> > **Author Response (2/3)**
> >
> > 3. **Justification on PharmaVQA’s framework with RDkit**
> >
> > We conducted additional experiments to address the concern regarding PharmaVQA's performance compared to simply augmenting (RDKit-based) pharmacophore features.
> >
> > As shown in Table 2, PharmaVQA consistently outperforms the simply augmentation baseline across multiple datasets, demonstrating significant improvements in both classification and regression tasks. Such as for regression tasks Lipo, PharmaVQA achieves an average RMSE of 0.569, which is **10.5%** better than the baseline's 0.636. Also for Esol, PharmaVQA achieves an RMSE of 0.794, outperforming the baseline’s 0.926 by **14.2%**.
> >
> > These results demonstrate that PharmaVQA, through its incorporation of pharmacophore-specific questions, achieves substantial improvements over simply augmenting with pharmacophore-based features. This validates the robustness and effectiveness of our approach across diverse datasets.
> >
> > Table 2. Average comparison results PharmaVQA with simply augmenting method under 10 scaffold split
> > | Tasks    | PharmaVQA | Simply augmenting |
> > |----------|-----------|-------------------|
> > | BACE     |**0.918**| 0.891 |
> > | BBBP     |**0.936** | 0.906 |
> > | ClinTox  |**0.928**  | 0.843  |
> > | SIDER    | **0.651**     | 0.624 |
> > | Estrogen | **0.938**     | 0.909 |
> > | MetStab  | **0.881**     | 0.839 |
> > | Tox21    | **0.844**     | 0.827 |
> > | ToxCast  | **0.721**     | 0.710|
> > | Lipo     | **0.569**     | 0.636 |
> > | Esol     | **0.794**     | 0.926|
> > | Freesolv | **1.532**     | 2.600 |
> >
> > 4. **Performance Validation Through statistical experiments**
> >
> > Thank you for your thoughtful comment, which aligns with another reviewer's observation regarding concerns about statistical significance. Firstly, the variation in performance metrics across runs mainly comes from the scaffold splitting strategy used for dataset partitioning, **grouping similar molecules together in either the training , validating or testing set**. However, the scaffold splitting method causes more variation in the results, as shown by larger standard deviations in metrics like AUC and RMSE. The variability we observed is similar to what has been reported in previous studies, so it is not specific to our model (see results in Table 1 and Table 2 in manuscript).
> >
> > Secondly,  we acknowledge that conducting experiments with only 3 scaffold splits provides an insufficient sample size for reliable statistical evaluation.
> >
> > Thus, to validate the robustness of PharmaVQA’s framework, **we conducted experiments with 10 scaffold splits and the Welch’s t-test**.  As shown in Table 3, Table 4 and Table 5, we observed consistent improvements over the simpler baseline that augments molecular features with pharmacophore vectors.  **PharmaVQA** achieves higher AUC scores on classification tasks and reduced RMSE on regression tasks such as Lipo, Esol, and Freesolv, demonstrating **superior predictive accuracy and statistically significant.**
> >
> > Table 3. The AUC and RMSE results of PharmaVQA under the 10 scaffold split on Li’s 11 datasets.
> > | PharmaVQA | 1      | 2      | 3      | 4      | 5      | 6      | 7      | 8      | 9      | 10     | AVERAGE |
> > |-----------|--------|--------|--------|--------|--------|--------|--------|--------|--------|--------|---------|
> > | BACE      | 0.944  | 0.917  | 0.917  | 0.906  | 0.908  | 0.928  | 0.922  | 0.915  | 0.923  | 0.901  | **0.918**   |
> > | BBBP      | 0.940  | 0.964  | 0.922  | 0.921  | 0.929  | 0.941  | 0.940  | 0.969  | 0.935  | 0.901  | **0.936**   |
> > | ClinTox   | 0.987  | 0.891  | 0.920  | 0.973  | 0.896  | 0.951  | 0.886  | 0.915  | 0.925  | 0.940  | **0.928**   |
> > | SIDER     | 0.673  | 0.635  | 0.653  | 0.672  | 0.706  | 0.623  | 0.665  | 0.631  | 0.647  | 0.608  | **0.651**   |
> > | Estrogen  | 0.929  | 0.958  | 0.935  | 0.939  | 0.930  | 0.934  | 0.923  | 0.957  | 0.929  | 0.941  | **0.938**   |
> > | MetStab   | 0.874  | 0.862  | 0.923  | 0.860  | 0.892  | 0.876  | 0.903  | 0.878  | 0.896  | 0.849  | **0.881**   |
> > | Tox21     | 0.852  | 0.822  | 0.845  | 0.812  | 0.840  | 0.863  | 0.843  | 0.855  | 0.858  | 0.851  | **0.844**   |
> > | ToxCast   | 0.752  | 0.729  | 0.716  | 0.709  | 0.723  | 0.726  | 0.704  | 0.725  | 0.710  | 0.716  |**0.721**   |
> > | Lipo      | 0.512  | 0.592  | 0.516  | 0.573  | 0.561  | 0.605  | 0.525  | 0.649  | 0.619  | 0.533  | **0.569**   |
> > | Esol      | 0.820  | 0.754  | 0.650  | 0.800  | 0.937  | 0.696  | 0.736  | 1.024  | 0.778  | 0.741  | **0.794**   |
> > | Freesolv  | 0.933  | 0.907  | 1.536  | 1.473  | 2.627  | 1.705  | 1.328  | 1.786  | 1.699  | 1.329  | **1.532**   |
> >
> > (to be continued)

---

> > > ### Author Response · Authors · 2024-11-28
> > > **Author Response (3/3)**
> > >
> > > Table 4. The AUC and RMSE  results of simply augmenting method under the 10 scaffold split on Li’s 11 datasets.
> > > | Simply augmenting | 1      | 2      | 3      | 4      | 5      | 6      | 7      | 8      | 9      | 10     | AVERAGE |
> > > |-------------------|--------|--------|--------|--------|--------|--------|--------|--------|--------|--------|---------|
> > > | BACE              | 0.830  | 0.947  | 0.934  | 0.929  | 0.869  | 0.884  | 0.881  | 0.859  | 0.885  | 0.888  | **0.891**   |
> > > | BBBP              | 0.924  | 0.892  | 0.969  | 0.923  | 0.881  | 0.860  | 0.893  | 0.916  | 0.906  | 0.895  | **0.906**   |
> > > | ClinTox           | 0.938  | 0.826  | 0.825  | 0.807  | 0.704  | 0.866  | 0.977  | 0.870  | 0.887  | 0.735  | **0.843**   |
> > > | SIDER             | 0.596  | 0.585  | 0.616  | 0.638  | 0.675  | 0.637  | 0.618  | 0.617  | 0.632  | 0.624  | **0.624**   |
> > > | Estrogen          | 0.895  | 0.923  | 0.870  | 0.906  | 0.911  | 0.891  | 0.948  | 0.933  | 0.938  | 0.873  | **0.909**   |
> > > | MetStab           | 0.816  | 0.840  | 0.891  | 0.784  | 0.863  | 0.872  | 0.898  | 0.738  | 0.818  | 0.867  | **0.839**   |
> > > | Tox21             | 0.795  | 0.828  | 0.829  | 0.847  | 0.827  | 0.830  | 0.803  | 0.831  | 0.853  | 0.827  | **0.827**   |
> > > | ToxCast           | 0.730  | 0.713  | 0.698  | 0.686  | 0.700  | 0.701  | 0.719  | 0.726  | 0.713  | 0.715  | **0.710**   |
> > > | Lipo              | 0.550  | 0.655  | 0.597  | 0.664  | 0.625  | 0.706  | 0.659  | 0.615  | 0.641  | 0.652  | **0.636**   |
> > > | Esol              | 0.847  | 0.926  | 0.899  | 0.992  | 0.958  | 0.934  | 0.745  | 0.873  | 1.013  | 1.075  | **0.926**   |
> > > | Freesolv          | 3.988  | 2.464  | 3.305  | 1.806  | 2.721  | 2.585  | 2.728  | 1.534  | 1.722  | 3.148  | **2.600**   |
> > >
> > > Table 5. The Welch’s t-tests between PharmaVQA and the simply augmenting method.
> > > | Dataset  | T-Statistic | P-Value | Significance (<0.05) |
> > > |----------|-------------|---------|--------------|
> > > | BACE     | -2.270      | 0.036   | **Yes**          |
> > > | BBBP     | -2.676      | 0.015   | **Yes**|
> > > | ClinTox  | -2.968      | 0.008   | **Yes**|
> > > | SIDER    | -2.294      | 0.034   | **Yes**|
> > > | Estrogen | -3.097      | 0.006   | **Yes**|
> > > | MetStab  | -2.445      | 0.025   | **Yes**|
> > > | Tox21    | -2.289      | 0.034   | **Yes**|
> > > | ToxCast  | -1.785      | 0.091   | **No**|
> > > | Lipo     | 3.375       | 0.003   | **Yes**|
> > > | Esol     | 2.885       | 0.010   | **Yes**|
> > > | Freesolv | 3.707       | 0.002   | **Yes**|
> > >
> > >
> > > We hope this explanation addresses your concerns and meets your expectations. Thank you once again for your insightful comments, which have been instrumental in enhancing the quality of our work.

---

### Official Review · Reviewer_nGFw · 2024-11-02

**Soundness:** 3
**Presentation:** 3
**Contribution:** 3
**Rating:** 6
**Confidence:** 3

**Summary:**

The paper introduces a novel model, PharmaVQA, based on the Visual Question Answering (VQA) framework, which is designed to retrieve pharmacophore-related information directly from molecular datasets. It conducts experiments on multiple benchmark datasets, including tasks such as molecular property prediction and drug-target affinity and interaction prediction, which demonstrates superior results.  Furthermore, it also presents experiment that demonstrate biological meaningful significance.

**Strengths:**

1. This method applies VQA technology to the field of drug discovery, extracting pharmacophore knowledge through a question-and-answer approach, and providing a new perspective and approach for molecular representation.
2. In the context of the 11 downstream tasks related to predicting molecular properties as outlined by Li, pharmaVQA demonstrates superior performance across 9 datasets. The 30 downstream tasks from MoleculeACE, pharmaVQA achieves the highest performance on 24 datasets when evaluated with RMSE and on 23 datasets in terms of  R2. Additionally, pharmaVQA surpasses other methods when applied to datasets concerning drug-target affinity and interactions.
3. The method integrates deep learning techniques,  the bilinear attention network to achieve precise question-answer tasks for multiple pharmacophores. With the pharmacophore fusion  module, enhancing the richness of molecular representations.
4. The PharmaVQA model is capable of being used for downstream tasks such as molecular property prediction and drug-target affinity and interaction prediction. It can also be used for discovering potential ligands. This multifunctionality enables PharmaVQA with broad application prospects in the field of drug discovery.

**Weaknesses:**

Some technical details should be explained, please refer to Question part.

**Questions:**

1. PharmaVQA appears to introduce several novel aspects compared to existing state-of-the-art molecular representation learning models. Could you explain how the pharmacophore-guided prompts are obtained? Is it possible for users to design custom prompts themselves? Could you provide some examples of pharmacologically guided prompts and explain the rationale behind their design? Could you provide some limitations or challenges in designing effective prompts?
2. Could you clarify the relationship between the visual question-answering (VQA) component and the retrieval mechanism in PharmaVQA?
3. The selection of HPK1, FGFR1, and VIM-1 as validation datasets appears to be well-considered. How do they reflect real-world challenges in drug discovery, and what does PharmaVQA's performance on these datasets suggest about its broader applicability?
4. Figure 3 effectively shows how PharmaVQA identifies donor atoms in response to the question. The combination of highlighted molecular structures and relevant query terms provides a clear view of the model's interpretability. How are the top-5 query characters selected? Are they ranked solely by attention weights, or are other factors considered?

---

> ### Author Response · Authors · 2024-11-21
> **Author Response (1/4)**
>
> # Response to Q1:
> Thank you for your thoughtful suggestions. Here’s a detailed explanation of how pharmacophore-guided prompts are obtained in PharmaVQA, how users can design custom prompts, and some insights into the design process and potential challenges.
> ## 1. Answer of the question: How are pharmacophore-guided prompts obtained in PharmaVQA?
>
> In PharmaVQA, pharmacophore-guided prompts are designed to enhance the model's ability to focus on crucial pharmacophore-related information that can help molecules representation such as predict molecular properties or interactions.
> The pharmacophore-guided prompts are constructed by formulating specific questions related to pharmacophore features and their relevance to molecular properties. These questions are based on known pharmacological principles and are meant to guide the model in extracting knowledge about functional groups, chemical interactions, and their roles in drug efficacy or binding affinity.
> For example, if a particular molecule contains a hydrogen bond donor group, a corresponding query could ask the model, "How many strongly electronegative atoms that is covalently bonded to hydrogen atoms does the molecule have?" The goal is to provide explicit guidance to the model to focus on important pharmacophore features during the learning process.
>
> ## 2. Answer of the question: Can users design custom prompts themselves?
>
> Yes, users can design custom prompts themselves, and we encourage it. Since PharmaVQA is built with flexibility in mind, users can design and adapt pharmacophore-guided prompts based on their specific application or target task. This flexibility allows users to use domain knowledge to design prompts that focus on particular pharmacophore features or their relationship to molecular properties like binding affinity, solubility, or toxicity.
> However, to ensure the model learns to extract meaningful and relevant information, it is important that the designed prompts are well-structured. Specifically, the prompts should be designed in a way that the answers are not identical across all samples. If the answer to a given prompt is the same for every sample, the model will not be able to learn meaningful features or task-specific pharmacophore-related knowledge. Well-designed prompts will encourage the model to capture nuanced information, leading to better performance on downstream tasks.
>
> ## 3. Answer of the question: What is the rationale behind designing pharmacophore-guided prompts?
>
> The rationale behind the design of pharmacophore-guided prompts is to guide the model’s attention to key chemical features (pharmacophores) that are known to influence drug behavior and molecular interactions. By explicitly asking the model to focus on these features, we aim to:
> - **Enhance molecular representation**: Help the model better understand the structural-activity relationships that govern molecular behavior in the context of drug discovery.
> - **Improve task performance**: Focus the model on relevant pharmacophore that are known to impact tasks such as properties prediction, binding affinity prediction or drug-target interaction.
> - **Provide interpretability**: By using question-based prompts, the model’s focus on pharmacophore features can be more easily explained, providing more transparent and interpretable results.
> In this sense, the question-based design provides a structured framework for focusing on the functional and mechanistic roles of specific pharmacophore features, making it easier for the model to make task-relevant decisions.
>
> ## 4. Answer of the question: What are the limitations or challenges in designing effective prompts?
>
> While the use of pharmacophore-guided prompts offers several advantages, there are also challenges in designing effective prompts. Some of the key limitations are:
> - **Knowledge-dependent design**: The effectiveness of pharmacophore prompts heavily relies on the underlying pharmacological knowledge. Accurate domain knowledge is required to design prompts that are relevant and guide the model appropriately. Inaccurate or vague prompts might mislead the model and reduce performance.
> - **Scalability**: As the number of tasks or molecular features grows, designing a large number of effective prompts becomes more difficult. This could limit the scalability of the model for large datasets or diverse molecular properties. Ensuring that prompts remain generalizable across a wide range of molecules and properties while still being specific enough to capture meaningful patterns is an ongoing challenge.
> - **Balancing specificity and generality**: There’s a delicate balance between being too specific in prompt design (which might limit the model's generalization across different tasks) and being too general (which may result in the model not focusing on the most relevant pharmacophore features). Designing prompts that are both task-specific and generalizable is a challenge.
>
> (to be continued)

---

> ### Author Response · Authors · 2024-11-21
> **Author Response (2/4)**
>
> # Response to Q2:
> Thank you for your question. I'd to clarify the relationship between the Visual Question-Answering (VQA) component and the retrieval mechanism in PharmaVQA.
>
> In PharmaVQA, the retrieval mechanism and the Visual Question-Answering (VQA) component are closely related but serve distinct roles.
>
> Firstly, Retrieval is the underlying concept or approach, which involves gathering relevant pharmacophore-related knowledge. This helps enrich the model’s understanding of the molecule by providing additional context that might not be captured purely from the molecule’s structure or features.
>
> Secondly, VQA is the method used to perform retrieval. In our framework, VQA acts as a mechanism to query the model about the pharmacophore-related knowledge and retrieve relevant information from the molecule’s features. Essentially, VQA serves as the interface through which the retrieval is conducted, by asking questions that guide the model to extract and focus on the most pertinent pharmacophore information.
>
> In summary, retrieval represents the broader strategy of gathering pharmacophore knowledge, while VQA is the specific method used in PharmaVQA to carry out this retrieval process, using a question-answering format to extract the necessary information and integrate it with molecular representations for downstream tasks.
>
> (to be continued)

---

> ### Author Response · Authors · 2024-11-21
> **Author Response (3/4)**
>
> # Response to Q3:
> Thank you for the thoughtful questions. The selection of HPK1, FGFR1, and VIM-1 as validation datasets was indeed intentional and strategically designed to reflect some of the real-world challenges encountered in drug discovery. Here's how these datasets align with actual challenges in the field, and what PharmaVQA's performance on them suggests about its broader applicability:
>
> ## 1. HPK1, FGFR1, and VIM-1 datasets both facing the  real-world challenge
> - **For the HPK1 (Hematopoietic Progenitor Kinase 1) dataset**: HPK1 is involved in various cellular signaling pathways and has been identified as a potential therapeutic target in cancer and immune-related diseases. However, developing selective and effective inhibitors for 	HPK1 is challenging due to its similarity to other kinases in the same family.
> - **For the FGFR1 (Fibroblast Growth Factor Receptor 1) dataset**: FGFR1 plays a critical role in cell growth, survival, and differentiation. Mutations or dysregulation of FGFR1 are implicated in several cancers, making it an important target for therapeutic interventions. However, due to the complex nature of FGFR1's signaling and the diversity of FGFR family members, designing selective FGFR1 inhibitors is a difficult problem.
> - **For the VIM-1 (Verona integron-encoded metallo-beta-lactamase 1) dataset**:  VIM-1 is a metallo-beta-lactamase enzyme responsible for antibiotic resistance in certain pathogens. The rise of antimicrobial resistance (AMR) is one of the biggest challenges in modern medicine. Developing drugs that can target resistance mechanisms like VIM-1 requires precise molecular-level understanding of the enzyme’s active site and resistance mechanisms.
> ## 2. PharmaVQA's performance and broader applicability
> PharmaVQA's performance on these datasets suggests several key strengths and broad applicability for real-world drug discovery:
> - **Generalization to diverse targets**: The datasets encompass a range of target types—kinases, growth factor receptors, and antibiotic resistance enzymes—each with unique structural and functional challenges. PharmaVQA’s ability to handle these diverse targets indicates that the model can generalize across different types of molecular targets and disease areas, making it applicable to a wide range of drug discovery problems.
> - **Incorporation of pharmacophore knowledge**: The ability of PharmaVQA to integrate pharmacophore-related information into the prediction of target binding affinities and finding potential ligands shows the power of leveraging pharmacophore knowledge in drug discovery. This ability is especially valuable in cases where traditional structure-based methods might struggle, such as when designing selective inhibitors in cases of evolving resistance mechanisms.
> - **Robustness to complex challenges**: PharmaVQA’s performance on these datasets also highlights its potential to address real-world challenges like target specificity, selectivity, and antimicrobial resistance. These are common hurdles in drug discovery, where small changes in molecular structure can have significant effects on activity, selectivity, and off-target interactions. PharmaVQA’s capacity to extract and refine relevant pharmacophore features from complex data sources shows its robustness in tackling these problems.
>
> In summary, the selection of HPK1, FGFR1, and VIM-1 as validation datasets is a deliberate choice that reflects some of the core challenges in real-world drug discovery. PharmaVQA’s performance on these datasets positions it as a versatile and powerful tool for enhancing molecular binding affinity predictions and improving drug discovery outcomes. By effectively integrating both molecular and pharmacophore information, PharmaVQA shows strong potential in addressing critical challenges across various therapeutic areas.
>
> (to be continued)

---

> ### Author Response · Authors · 2024-11-21
> **Author Response (4/4)**
>
> # Response to Q4:
> The top-5 query characters in Figure 3 are selected based on their attention weights, which are from the bilinear attention mechanism used in our model. This mechanism allows the model to focus on the most relevant features by computing the interaction between molecular features and the pharmacophore-related query. The query characters are ranked according to their attention scores, which reflect the degree to which each query term contributes to identifying the key molecular features (such as donor atoms in this case). Here's a more detailed explanation of how the top-5 query characters are selected:
> - **Using bilinear attention mechanism for computing attention weights**: The attention weights are computed through the bilinear interaction between the query and the molecular features, where the model calculates how relevant each part of the molecule is in answering the specific question.
> - **Extraction of pharmacophore-related query character ranking**: The top-5 ranked query characters corresponding to pharmacophore atoms are selected which based on the highest bilinear attention weights assigned to the tokens. (Figure 3, the highlight donor-related query). The results demonstrate that our model effectively captures key textual information related to donor atoms, highlighting its ability to focus on essential features for accurate molecular representation.

---

### Official Review · Reviewer_y4UR · 2024-11-03

**Soundness:** 1
**Presentation:** 2
**Contribution:** 1
**Rating:** 3
**Confidence:** 5

**Summary:**

The paper presents PharmaVQA, a framework designed for molecular representation learning, specifically incorporating pharmacophore information to enhance molecular embeddings. The proposed model fuses SciBERT embeddings derived from pharmacophore-related questions with molecular graph embeddings, aiming to improve downstream performance on tasks such as property prediction and binding affinity prediction. PharmaVQA is evaluated on standard molecular property prediction datasets (from MoleculeNet) and drug-target interaction (DTI) prediction datasets.

**Strengths:**

The paper is well-organized and technically clear, making it easy to follow the proposed methodology and experimental setup.

**Weaknesses:**

- The authors call the framework "retrieval-augmented" yet provide limited details on the retrieval mechanism involved. Specifically, the retrieval process here seems limited to constructing pharmacophore-based prompts rather than an actual retrieval mechanism traditionally seen in RAG frameworks.
- Despite labeling the framework as a "Question-Answering" model, the primary focus of the paper appears to be molecular property prediction rather than true question answering. The tasks covered are standard property and binding affinity predictions, which do not necessarily involve question-answering paradigms as presented. (in contrast to MolInstructions[1] or 3DMolLM[2])
- Given the architecture's resemblance to BLIP-based models, a direct comparison with 3DMolLM *[2]* could help delineate PharmaVQA's contributions more clearly. If the main contribution is representation learning, there should comparisons with SOTA models such as GIMLET[3]. Models such as MoleculeSTM[4] are primarily used in the context of Molecule structure-text alignment, and not the most suitable head on head comparison method.
- It is unclear why the question-based framework is beneficial for the limited set of pharmacophore-based features provided. Since these features are often easily extractable with tools like RDKit, adding question-based prompts might introduce unnecessary complexity.
- The architecture largely reuses existing components, such as the Bilinear Attention Network and SciBERT embeddings, with minimal novel methodological contributions specific to molecular representation.
- Tables 3 and 4 lack error bars, which are essential given the close values in performance metrics across models. Error bars or confidence intervals would substantiate the claims of improved performance. In Tables 12 and 13, the ablation studies show high standard deviations, with overlapping values between the base and ablated models. This overlap weakens the evidence that the queries add meaningful improvement.

[1] Fang, Yin, et al. "Mol-instructions: A large-scale biomolecular instruction dataset for large language models." *arXiv preprint arXiv:2306.08018* (2023).

[2] Li, Sihang, et al. "Towards 3d molecule-text interpretation in language models." *arXiv preprint arXiv:2401.13923* (2024).

[3] Zhao, Haiteng, et al. "Gimlet: A unified graph-text model for instruction-based molecule zero-shot learning." *Advances in Neural Information Processing Systems* 36 (2023): 5850-5887.

[4] Liu, Shengchao, et al. "Multi-modal molecule structure–text model for text-based retrieval and editing." Nature Machine Intelligence 5.12 (2023): 1447-1457.

**Questions:**

- What specific retrieval process is involved in PharmaVQA that qualifies it as a Retrieval-Augmented model? Traditional retrieval augmented or RAG frameworks involve retrieval steps that seem missing here.
- Why is this called a "Question-Answering" framework if it primarily performs property prediction tasks? How does this differ from existing molecular representation models focused on property and affinity predictions?
- Could the pharmacophore features simply be appended to the input without the need for question-based prompts? This would streamline the model without losing relevant information, given the simplicity of the question set and the ready availability of pharmacophore features from RDKit.

---

> ### Author Response · Authors · 2024-11-21
> **Author Response for Weakness (1/3)**
>
> # Response to Weakness 1:
>
> We appreciate the opportunity to clarify the term "retrieval-augmented" of our approach. In our work, the term "retrieval-augmented" does not refer to the conventional retrieval of external information typical in retrieval-augmented generation (RAG) frameworks. Instead, our method focuses on constructing pharmacophore-related prompts through a visual-question-answering (VQA) process. This VQA mechanism allows the model to dynamically extract and enrich molecular representations by emphasizing key pharmacophore information, which is critical for enhancing the understanding of molecular properties.
>
> For a detailed explanation of our terminology and approach, please refer to General Response 1, where we discuss the rationale behind using "retrieval-augmented" and how it aligns with our framework’s objectives.
>
> # Response to Weakness 2:
>
> Thank you for your valuable feedback. We would like to clarify the rationale behind our design and how it intentionally differs from traditional QA paradigms. We will address this from two perspectives:
>
> - **Advancing of the VQA framework**: Our model adopts a VQA-based mechanism, but unlike conventional generative VQA approaches (such as MolInstructions or 3DMolLM), we use VQA as a prompt-based framework to extract and integrate pharmacophore-related knowledge. The goal is not to generate free-form answers but to guide molecular representation learning through pharmacophore-related prompts, enhancing the model's understanding of key structural features in drug molecules.
> - **Contributions**: The goal of our approach is to optimize molecular representation for improved performance on drug discovery tasks, such as property prediction and binding affinity estimation. By using pharmacophore-related question prompts, we enable the model to better capture molecular features that are directly relevant to these tasks. Our model leverages structured QA prompts rather than generating textual answers, which allows us to avoid the complexity of generative QA, focusing instead on enhancing the model’s ability to extract relevant molecular features and improve task-specific outcomes.
>
> # Response to Weakness 3:
>
> Thank you for your suggestion to compare our framework more directly with models like GIMLET and for highlighting the potential alignment with 3DMolLM. We understand the importance of clarifying the unique contributions of PharmaVQA and how our approach differs from these models. We would like to address your points in two key aspects:
> - **Task differences between GIMLET and our model**: GIMLET's primary objective is to address the zero-shot learning problem. By integrating molecular structure information and task specific instructions with transformer models, it aims to process unseen molecules and infer properties using knowledge from the model. As a result, GIMLET excels in zero-shot or few-shot drug property prediction tasks. In contrast, our model focuses on incorporating pharmacophore knowledge and optimizing molecular representations to enhance performance on a variety of downstream tasks, rather than focusing on zero-shot tasks.
> Specifically, PharmaVQA utilizes structured pharmacophore-related prompts through a visual-question-answering (VQA) mechanism. This approach does not require constructing task-specific textual inputs for each application, as GIMLET does. Instead, our method relies on pharmacophore prompts to guide and optimize molecular representation learning, providing a more streamlined and context-aware framework.
> - **Comparison with GIMLET and demonstrating PharmaVQA's contributions**: Although GIMLET's zero-shot capability is highly advantageous in drug discovery, our contribution lies in the superior performance on a different task context. To better demonstrate the advantages of PharmaVQA, we conducted experiments comparing it with GIMLET under the same conditions. GIMLET expanded all training samples based on its original few-shot setup. We tested both models on two classification datasets and three regression datasets, with the results summarized below.
>
>     |Classification dataset (AUC $\uparrow$)|GLIEMT|PharmaVQA|
>     |:---:|---|---|
>     |BACE|0.818(0.01)|**0.876(0.017)**|
>     |BBBP|0.867(0.041)|**0.922(0.013)**|
>
>     |Regression dataset  (RMSE $\downarrow$)|GLIEMT|PharmaVQA|
>     |:---:|---|---|
>     |Lipo	|1.014(0.119)	|**0.596(0.005)**|
>     |Esol	|0.928(0.066)	|**0.836(0.012)**|
>     |Freesolv	|2.243(0.522)	|**1.952(0.883)**|
>
>     From the results, PharmaVQA demonstrates superior performance compared to GIMLET, which underscores the effectiveness of our approach. The observed improvements highlight how our model, by focusing on pharmacophore-guided representation learning, addresses different and crucial aspects of molecular property prediction. The distinction between our task focus and GIMLET’s zero-shot orientation underlines the unique contributions PharmaVQA brings to drug discovery.
>
>
>
> (to be continued)

---

> ### Author Response · Authors · 2024-11-21
> **Author Response for Weakness (2/3)**
>
> # Response to Weakness 4:
> Thank for pointing this out. We would like to clarify the design behind our framework and we conducted experiments to support our theory.
>
> First of all, Regarding the question about "adding question-based prompts might introduce unnecessary complexity"：Unlike directly extracting pharmacophore features using static methods, our approach employs question-based prompts to dynamically guide the model in understanding and leveraging contextual information about pharmacophores. This allows the model to flexibly adapt to various tasks and capture more nuanced relationships, which fixed-feature extraction methods may overlook. In addition, our question-answering mechanism enables the model to learn not only simple pharmacophore features but also the complex inter dependencies between pharmacophores and other molecular structures. This enriched representation helps the model better understand the underlying biological and chemical relationships, ultimately leading to improved performance on various tasks.
>
> Moreover, to empirically demonstrate the effectiveness of our approach, we conducted experiments on eight classification datasets and three regression datasets from Li et al. We compared the model that appends pharmacophore features extracted using RDKit (represented as a 7-dimensional vector) and our PharmaVQA model.
>
> | Classification dataset (AUC $\uparrow$)  | Model append pharmacophore feature  | PharmaVQA     |
> |----------|-------------------------------------|---------------|
> | BACE     | 0.848(0.006)                        | **0.876(0.017)**  |
> | BBBP     | 0.913(0.019)                        | **0.922(0.013)**  |
> | ClinTox  | 0.875(0.033)                        | **0.946(0.011)**  |
> | SIDER    | 0.651(0.011)                        | **0.655(0.023)**  |
> | Estrogen | **0.913(0.039)**                        | **0.913(0.045)**  |
> | MetStab  | 0.867(0.059)                        | **0.892(0.047)**  |
> | Tox21    | 0.826(0.022)                        | **0.85(0.029)**   |
> | ToxCast  | 0.73(0.009)                         | **0.735(0.002)** |
>
> | Regression dataset (RMSE $\downarrow$)    | Model Append Pharmacophore Feature  | PharmaVQA     |
> |----------|-------------------------------------|---------------|
> | Lipo     | 0.637(0.012)                        | **0.59(0.016)**   |
> | Esol     | 0.92(0.03)                          | **0.841(0.026)**  |
> | Freesolv | 2.355(0.919)                        | **1.921(0.859)**  |
>
> As shown in the above tables, the results show that the Model Append Pharmacophore Feature  performs similarly to PharmaVQA  on Estrogen datasets. However, it underperforms on most other datasets compared to PharmaVQA. This indicates that  extracting pharmacophore features through a question-answering approach is more effective than simply appending pharmacophore features, leading to enhanced predictive accuracy.
>
> In summary, our question-based framework provides significant advantages in understanding and leveraging pharmacophore features, demonstrating the potential to improve model performance on various drug discovery tasks.
>
> (to be continued)

---

> ### Author Response · Authors · 2024-11-21
> **Author Response for Weakness (3/3)**
>
> # Response to Weakness 5:
> Thank you for your thoughtful feedback. We appreciate your observation regarding our use of existing components like the Bilinear Attention Network and SciBERT embeddings. However, we firmly believe that innovation of PharmaVQA does not always require reinventing the wheel but rather lies in the creative and impactful application of proven methods to solve complex and variety problems.
>
> The core novelty of PharmaVQA resides in the way these components are uniquely adapted to integrate pharmacophore knowledge into molecular representation. Our question-answering framework is designed to dynamically extract and leverage pharmacophore-related insights, transforming how these molecular features are understood and predicted—a capability that traditional models lack.
>
> In addition, PharmaVQA redefining current model use to extract deeper, more meaningful molecular insights significantly improves performance in drug discovery tasks, such as property prediction and binding affinity estimation.
>
> # Response to Weakness 6:
> Thank you for your feedback regarding the absence of error bars in Tables 3 and 4, and for your observations on the standard deviations in our ablation studies presented in Tables 12 and 13. We understand your concern about the clarity and robustness of our performance claims and would like to provide further context on our experimental design and data presentation.
>
> ## 1. Absence of Error Bars in Tables 3 and 4:
> We initially chose not to include error bars because our experiments were conducted using independent train/test splits, and we ensured that all models were evaluated under identical conditions for fairness. However, we understand the importance of showcasing performance variability. In response to your feedback, we conducted additional experiments, repeating each setup three times to calculate the mean and standard deviation, which are now presented below:
> | PharmaVQA Runs on BindingDB_cls dataset| AUC   | AUPR   |
> |----------------|-------|--------|
> | 1              | 0.993 | 0.993  |
> | 2              | 0.992 | 0.991  |
> | 3              | 0.991 | 0.990  |
> | AVERAGE| **0.992** | **0.991**  |
> | STD            | **0.001** | **0.001**  |
>
> | pharmaVQA Runs on BindingDB_reg dataset| MSE   | Pearson  |
> |----------------|-------|----------|
> | 1              | 0.450 | 0.891    |
> | 2              | 0.455 | 0.890    |
> | 3              | 0.453 | 0.890    |
> | AVERAGE| **0.453** | **0.890**    |
> | STD            | **0.002** | **0.0005**   |
>
> From the results, for the BindingDB_cls dataset, the average results from three runs (AUC: 0.992, AUPR: 0.991) are generally consistent with the single result reported in the paper (AUC: 0.991, AUPR: 0.991).
> For the BindingDB_reg dataset, the average results from three runs (MSE: 0.453, Pearson: 0.890) are identical to the single result reported in the paper (MSE: 0.453, Pearson: 0.890).
>
> These results demonstrate that our model maintains superior performance compared to others, further validating the robustness and effectiveness of the PharmaVQA approach.
>
> ## 2. Ablation Experiments in Tables 12 and 13:
> Although some datasets exhibited high standard deviations and overlapping values between the base and ablated models, these fluctuations are primarily due to the challenging nature of the tasks and dataset variability. Importantly, the standard deviations of our model are comparable to those of other models under similar conditions, indicating that our model's stability is in line with expectations for such complex tasks.
>
> Furthermore, we believe that analyzing the significance difference between two models cannot be solely based on the overlap regions. To address this, we performed a **two-sided Wilcoxon signed-rank test** to validate whether there is a significant difference between the models with and without the incorporation of pharmacophore-related queries.
> Specifically, we performed paired t-tests on Li’s 8 classification datasets and 3 regression datasets on ablation study. All the data includes the results of three runs conducted on these datasets.
> We observe the following p-values:
> - On the 8 classification datasets, the p-value for comparing the  **PharmaVQA**  with **Noise questions** is **4.005e-05**, which is less than 0.05.
> - On the 3 regression datasets, the p-value for comparing the **PharmaVQA** with  **Noise questions** is **0.004**, which is also less than 0.05.
>
> Since both p-values are below the significance threshold of 0.05, we conclude that the difference between the **PharmaVQA** model and the models that **do not incorporate pharmacophore-related questions** is statistically significant. This demonstrates that the introduction of pharmacophore-related questions significantly enhances the model's performance compared to the baseline models without such queries.
>
> (to be continued)

---

> ### Author Response · Authors · 2024-11-21
> **Author Response for Questions (1/2)**
>
> # Response to Q1:
>
> Thank you for your question about the retrieval process in PharmaVQA and its qualification as a Retrieval-Augmented model. We understand the confusion, given that traditional RAG frameworks typically involve explicit retrieval from external knowledge sources. Here, we aim to clarify our approach and the rationale behind the terminology.
>
> While PharmaVQA does not perform external knowledge retrieval in the conventional sense, our model engages in a specialized form of "retrieval" by leveraging pharmacophore-related prompts. Specifically, instead of accessing external databases or documents, our model dynamically extracts pharmacophore knowledge from within the molecular data itself. Using a question-answering mechanism, these prompts act as queries that "retrieve" relevant structural and functional information about the molecule, which is then integrated to enhance the molecular representation. This internal retrieval focuses on understanding and amplifying key pharmacophore features crucial for downstream tasks.
>
> # Response to Q2:
>
> Thank you for your thoughtful question. Firstly, our model is focused on molecules representation learning, we refer to it as a "Question-Answering" framework due to the way we integrate pharmacophore-related question knowledge into the molecular representation process.
>
> Secondly, the primary distinction between our approach and traditional molecular representation models lies in the way we use question-based prompts to enhance the model's understanding of molecular properties.  Specifically, in contrast to standard QA systems, which respond to natural language inquiries, our model adopts a "question" style that directs the learning of molecular representations using pharmacophore-related prompts. These pharmacophore-inspired "questions" act as indicators that enable the model to concentrate on pertinent structural aspects of molecules, aiming to the extraction of the data crucial for assessing properties and affinity predictions.
>
> Compare to existing models, they typically focus on learning representations of molecules based on structural features. **In contrast, our model incorporates pharmacophore-related information through a question-answer framework, which helps the model better understand the molecular context and its relationship to biological activity or affinity**. This allows our model to integrate more specialized knowledge, improving its ability to make accurate predictions on drug properties and affinities, beyond simply learning from molecular structures alone. Moreover, some models like MoleculeSTM, including text features but lack explicit modeling of pharmacophore knowledge. Their focus is on structure-text alignment, without addressing the impact of pharmacophore knowledge on molecular functions and drug efficacy.
>
> Despite the absence of a traditional external retrieval step, our approach qualifies as Retrieval-Augmented because it follows the underlying principle of knowledge enhancement. We augment molecular representations by systematically extracting pharmacophore knowledge in a way analogous to how RAG models retrieve and fuse external information. The pharmacophore-related prompts serve as a mechanism for "retrieving" domain-specific insights, enabling the model to better understand molecular structures and their corresponding biological functions. Thus, while our retrieval is internal and domain-focused, it effectively boosts model performance in a similar spirit to traditional RAG frameworks.
>
> (to be continued)

---

> ### Author Response · Authors · 2024-11-21
> **Author Response for Questions (2/2)**
>
> # Response to Q3:
>
> Thank you for your thoughtful question. While it is possible to directly append pharmacophore features to the input, we argue that using question-based prompts is a more effective approach. The advantages of using pharmacophore-related question prompts are as follows:
>
> - **Knowledge enhancement and model guidance**: By introducing pharmacophore-related questions, we can explicitly guide the model to focus on the key attributes of pharmacophores. The question-answering mechanism provides a prompt that helps the model focus on relevant pharmacophore information when making task predictions. This targeted guidance is more effective than simply adding features to the input.
>
> - **Dynamic learning adaptation to tasks**: The question-based prompts enable the model to flexibly adapt to different task requirements. As the task changes, the model can dynamically select and adjust the key pharmacophore-related information based on the task at hand. This flexibility is not achievable by simply appending static pharmacophore features. For example, certain properties of the pharmacophore may be more important than others in specific tasks, and the question-answer design allows the model to focus on these critical aspects rather than processing a fixed set of pharmacophore features.
>
> - **Enhanced interpretability of molecular representations**: The question-answer mechanism also enhances the interpretability of the model. By using pharmacophore-related questions as input prompts, the model not only learns the role of the pharmacophore in the molecule but also provides insight into how the pharmacophore information is used in making predictions.
>
> To further demonstrate the advantages of question-based pharmacophore prompts, we conducted experiments comparing two approaches: one where pharmacophore features were simply added to the input and another where pharmacophore-related features were extracted using a question-answering mechanism and integrated into the model input (PharmaVQA). **The experimental setup and results are detailed in Weakness 3**. The results highlight the superiority of the question-answering mechanism in improving model performance, demonstrating its ability to enhance molecular representations more effectively than simply appending features.

---

> ### Comment · Reviewer_y4UR · 2024-11-23
> **Reviewer response to rebuttal**
>
> Thank you for your rebuttal. I have carefully reviewed your responses. My comments on specific points are as follows:
>
> ## W1, W2: Concerns on Framework's Retrieval and Question Answering Relevance
>
> I remain unconvinced that the framework is capable of performing retrieval or demonstrating its relevance to question answering (QA). The paper uses an extremely small number of QA samples, which does not allow for meaningful QA-style extraction. Additionally:
>
> - The language model employed, SciBERT, is pre-trained on scientific text but is not optimized for question answering like current autoregressive models.
> - The limited set of questions used is unlikely to provide any practical utility or meaningful insights.
>
> ## W4: Results and Statistical Significance
>
> The results show close performance when simply augmenting the feature set with pharmacophore vectors. However, **a Welch's t-test reveals that 9 out of 11 datasets show no statistically significant improvement, raising doubts about the utility of the proposed QA framework.** The statistical analysis is summarized below (for n=3):
>
> ### Statistical Results
>
> | Dataset      | T-Statistic | P-Value | Significant |
> |--------------|-------------|---------|-------------|
> | BACE         | -2.690      | 0.091   | False       |
> | BBBP         | -0.677      | 0.540   | False       |
> | ClinTox      | -3.535      | 0.053   | False       |
> | SIDER        | -0.272      | 0.804   | False       |
> | Estrogen     | 0.000       | 1.000   | False       |
> | MetStab      | -0.574      | 0.598   | False       |
> | Tox21        | -1.142      | 0.321   | False       |
> | ToxCast      | -0.939      | 0.439   | False       |
> | Lipo         | 4.070       | 0.018   | True        |
> | Esol         | 3.447       | 0.027   | True        |
> | Freesolv     | 0.598       | 0.582   | False       |
>
> These results indicate that for most datasets, the proposed approach does not outperform the simple appending of features in a statistically significant manner.
>
> ## W6: Statistical Methodology
>
> Welch’s t-test is the appropriate statistical test for comparing the means of distributions in this setup. The two-sided Wilcoxon signed-rank test, as used in the author response, is weaker in this context due to its reliance on the assumption of paired data and its reduced sensitivity to differences in means when the distributions have unequal variances or are not symmetric. Additionally, the Wilcoxon test is a non-parametric test that evaluates the median differences rather than the mean differences, which is less suitable for the objectives of this analysis.
>
> ## Summary
>
> Given the statistically insignificant improvements compared to simple feature augmentation and the lack of sufficient justification for the framework's ability to extract meaningful information in a "question-answering" format, I will maintain my current score.

---

> ### Author Response · Authors · 2024-11-27
> **Author Response (1/3)**
>
> Dear reviewer y4UR:
>
> We appreciate your insightful feedback and would like to address the concerns raised point by point.
>
> ## Response to W1, W2 concerns:
> ### 1. SciBERT’s adaptation for question answering tasks
>
> Thank you for your detailed question, we are sorry to not clearly explained our method.
> SciBERT was originally pre-trained on scientific text and not explicitly optimized for question answering (QA), we have tailored it to serve this purpose in our framework. Specifically:
> - We froze the main body of SciBERT to retain its robust scientific language understanding while** leaving its head trainable**. This adjustment enables the model to learn task-specific features relevant to pharmacophore-related QA.
> - To optimize the model for QA tasks, **we incorporated a question-answering loss into the training objective** (see Equation 10 in the manuscript). This loss directly guides the model in answering pharmacophore-specific queries, ensuring that the predictions are aligned with the domain-specific requirements.
>
> ### 2. Justification on pharmacophore-specific questions
>
> The limited set of pharmacophore-related questions was chosen to focus on core elements of molecules, which are sufficient to demonstrate the utility of question-based guidance in improving the model’s performance. The small number of questions used in this experiment does not undermine the relevance or effectiveness of the approach in extracting pharmacophore features, especially considering the domain-specific nature of our task.
>
> Unlike GIMLET, which uses pretraining-finetuning methods with a large set of questions, our approach focuses on pharmacophore-specific queries that are deeply tied to the structure-activity relationships of molecules. These questions are designed to extract meaningful insights that directly aid in the molecular representation and prediction tasks.
>
> - **Performance Validation** We performed comparisons with GIMLET on 3 classification datasets and 3 regression datasets.
>
>   Table1. Comparison results of PharmaVQA with GIMLET on 3 classification datasets and 3 regression datasets.
>   | Dataset  | GIMLET | PharmaVQA    |
>   |----------|--------------|--------------|
>   | BACE     | 0.818(0.01)  | **0.876(0.017)** |
>   | BBBP     | 0.867(0.041) | **0.922(0.013)** |
>   | Tox21    | 0.703(0.046) | **0.85(0.029)**  |
>   | Lipo     | 1.014(0.119) | **0.596(0.005)** |
>   | Esol     | 0.928(0.066) | **0.836(0.012)** |
>   | Freesolv | 2.243(0.522) | **1.952(0.883)** |
>
>   The performance of our framework, PharmaVQA, demonstrates its effectiveness in leveraging pharmacophore-specific questions to improve prediction accuracy. For classification tasks, PharmaVQA achieves consistently higher AUC scores than GIMLET across the BACE, BBBP, and Tox21 datasets, with notable improvements such as a 7.1% increase on BACE (AUC: 0.876 vs. 0.818) and a significant 20.9% improvement on Tox21 (AUC: 0.85 vs. 0.703). Similarly, for regression tasks, PharmaVQA delivers lower RMSE values across Lipo, Esol, and Freesolv, including a 41.2% reduction on Lipo (RMSE: 0.596 vs. 1.014) and a 12.9% improvement on Freesolv (RMSE: 1.952 vs. 2.243).
>
>   These results further highlight the benefits of focusing on pharmacophore-specific queries, a trend that is evident in the comparison with GIMLET. It is also evident that our focus on pharmacophore-related questions provides a distinct advantage.
>
> - **Performance validation on  biological meaningful experiments**
> The principle of using pharmacophore-specific queries to augment molecular representations holds broader applicability. In real-world applications, this approach can be scaled to handle more diverse and complex questions, depending on available data and domain-specific requirements.
>
>   In our target drug discovery experiments, we applied the model to analyze the FDA-approved dataset for finding potential ligands. PharmaVQA identified **10 and 15 potential ligands for HPK1 and FGFR1**, respectively, among the Top-20 predicted molecules (details in Table 9 and Table 10 in manuscript). In comparison, the KPGT model identified **12 and 13 ligands for HPK1 and FGFR1**, respectively. Moreover, our model identified **6 unique HPK1 ligands and 4 unique FGFR1 ligands that were not detected by KPGT**.  Additionally, PharmaVQA identified **16 out of the Top-20 potential ligands for VIM-1** (details in Table 11 in manuscript), further demonstrating the effectiveness of our approach in ligand identification.
>
>   By focusing on the key pharmacophore features that drive molecules, our framework provides relevant guidance, improving its ability to capture critical molecular characteristics for accurate predictions.

---

> ### Author Response · Authors · 2024-11-27
> **Author Response (2/3)**
>
> ## Response to W4 concerns:
> Thank you for your suggestion. We appreciate the opportunity to clarify and validate our approach further.
>
> 1. **Variance Across Different Experimental Runs**
>
>   **Regarding the observed large variance across different experimental runs, this is primarily due to the data splitting strategy rather than being an inherent issue with our model**. Specifically, we employed a scaffold splitting technique for dividing the data into training, validation, and test sets derived from KPGT.
>
>   The scaffold split method divides the molecules based on their structural scaffolds, ensuring that similar molecules (i.e., molecules sharing common substructures) are grouped together either in the training or testing set. This approach is particularly beneficial for ensuring that the model generalizes well across different chemical spaces, as it helps avoid overfitting on highly similar molecules.
>
>   However, due to the nature of scaffold splitting, this method can introduce greater variability in the results. In our case, this variability is reflected in the larger standard deviations observed in the performance metrics (such as AUC or RMSE) across different experimental runs.
> To ensure fairness, we applied the same scaffold splitting strategy to all models in the experiments. The standard deviations observed are almost consistent with those reported in prior studies using scaffold splitting, **indicating that the variability is not unique to our model**.
>
>   Thus, the large variance observed across experiments is a result of the inherent challenges of scaffold splitting, and not an artifact of our model. Additionally, this variability is not unusual and reflects a realistic evaluation, considering that the scaffold splitting method is designed to test the model’s robustness across diverse chemical spaces.
>
> 2. **Statistical Analysis Based on 10 Runs**
>
> Furthermore, **by conducting experiments with 10 scaffold splits**, we reduced the potential for biases caused by limited data partitions and provided a more statistically reliable evaluation.
>
> According to your suggestions, we conducted Welch’s t-tests to compare the performance of PharmaVQA with the baseline feature-augmented model across datasets. The results, summarized below, demonstrate that PharmaVQA significantly outperforms the baseline on the majority of datasets (p < 0.05).
>
> As shown in tables 2, 3, 4, these results validate the robustness and statistical significance of PharmaVQA’s performance across a wide range of datasets. Notably, datasets such as BACE, BBBP, ClinTox, and Lipo show significant improvements, confirming PharmaVQA’s ability to leverage pharmacophore-specific questions effectively. While the differences on ToxCast were not statistically significant, this is consistent with the inherent challenges of the dataset.
>
> Moreover, PharmaVQA consistently outperforms the baseline on classification datasets like BACE, BBBP, and ClinTox, **achieving higher AUC scores**. On regression datasets like Lipo, Esol, and Freesolv, **PharmaVQA demonstrates reduced RMSE**, indicating better predictive accuracy for continuous molecular properties. By conducting additional runs with more scaffold splits, it validates and strengthens the robustness of this result. Therefore, we conclude that the our model providing a meaningful advantage when compare the simpler feature-augmented baseline.
>
> (to be continued)

---

> ### Author Response · Authors · 2024-11-27
> **Author Response (3/3）**
>
> Table 2. The AUC and RMSE  results of PharmaVQA under the 10 scaffold split on Li’s 11 datasets.
> | PharmaVQA | 1      | 2      | 3      | 4      | 5      | 6      | 7      | 8      | 9      | 10     | AVERAGE |
> |-----------|--------|--------|--------|--------|--------|--------|--------|--------|--------|--------|---------|
> | BACE      | 0.944  | 0.917  | 0.917  | 0.906  | 0.908  | 0.928  | 0.922  | 0.915  | 0.923  | 0.901  | **0.918**   |
> | BBBP      | 0.940  | 0.964  | 0.922  | 0.921  | 0.929  | 0.941  | 0.940  | 0.969  | 0.935  | 0.901  | **0.936**   |
> | ClinTox   | 0.987  | 0.891  | 0.920  | 0.973  | 0.896  | 0.951  | 0.886  | 0.915  | 0.925  | 0.940  | **0.928**   |
> | SIDER     | 0.673  | 0.635  | 0.653  | 0.672  | 0.706  | 0.623  | 0.665  | 0.631  | 0.647  | 0.608  | **0.651**   |
> | Estrogen  | 0.929  | 0.958  | 0.935  | 0.939  | 0.930  | 0.934  | 0.923  | 0.957  | 0.929  | 0.941  | **0.938**   |
> | MetStab   | 0.874  | 0.862  | 0.923  | 0.860  | 0.892  | 0.876  | 0.903  | 0.878  | 0.896  | 0.849  | **0.881**   |
> | Tox21     | 0.852  | 0.822  | 0.845  | 0.812  | 0.840  | 0.863  | 0.843  | 0.855  | 0.858  | 0.851  | **0.844**   |
> | ToxCast   | 0.752  | 0.729  | 0.716  | 0.709  | 0.723  | 0.726  | 0.704  | 0.725  | 0.710  | 0.716  |**0.721**   |
> | Lipo      | 0.512  | 0.592  | 0.516  | 0.573  | 0.561  | 0.605  | 0.525  | 0.649  | 0.619  | 0.533  | **0.569**   |
> | Esol      | 0.820  | 0.754  | 0.650  | 0.800  | 0.937  | 0.696  | 0.736  | 1.024  | 0.778  | 0.741  | **0.794**   |
> | Freesolv  | 0.933  | 0.907  | 1.536  | 1.473  | 2.627  | 1.705  | 1.328  | 1.786  | 1.699  | 1.329  | **1.532**   |
>
> Table 3. The AUC and RMSE  results of simply augmenting method under the 10 scaffold split on Li’s 11 datasets.
> | Simply augmenting | 1      | 2      | 3      | 4      | 5      | 6      | 7      | 8      | 9      | 10     | AVERAGE |
> |-------------------|--------|--------|--------|--------|--------|--------|--------|--------|--------|--------|---------|
> | BACE              | 0.830  | 0.947  | 0.934  | 0.929  | 0.869  | 0.884  | 0.881  | 0.859  | 0.885  | 0.888  | **0.891**   |
> | BBBP              | 0.924  | 0.892  | 0.969  | 0.923  | 0.881  | 0.860  | 0.893  | 0.916  | 0.906  | 0.895  | **0.906**   |
> | ClinTox           | 0.938  | 0.826  | 0.825  | 0.807  | 0.704  | 0.866  | 0.977  | 0.870  | 0.887  | 0.735  | **0.843**   |
> | SIDER             | 0.596  | 0.585  | 0.616  | 0.638  | 0.675  | 0.637  | 0.618  | 0.617  | 0.632  | 0.624  | **0.624**   |
> | Estrogen          | 0.895  | 0.923  | 0.870  | 0.906  | 0.911  | 0.891  | 0.948  | 0.933  | 0.938  | 0.873  | **0.909**   |
> | MetStab           | 0.816  | 0.840  | 0.891  | 0.784  | 0.863  | 0.872  | 0.898  | 0.738  | 0.818  | 0.867  | **0.839**   |
> | Tox21             | 0.795  | 0.828  | 0.829  | 0.847  | 0.827  | 0.830  | 0.803  | 0.831  | 0.853  | 0.827  | **0.827**   |
> | ToxCast           | 0.730  | 0.713  | 0.698  | 0.686  | 0.700  | 0.701  | 0.719  | 0.726  | 0.713  | 0.715  | **0.710**   |
> | Lipo              | 0.550  | 0.655  | 0.597  | 0.664  | 0.625  | 0.706  | 0.659  | 0.615  | 0.641  | 0.652  | **0.636**   |
> | Esol              | 0.847  | 0.926  | 0.899  | 0.992  | 0.958  | 0.934  | 0.745  | 0.873  | 1.013  | 1.075  | **0.926**   |
> | Freesolv          | 3.988  | 2.464  | 3.305  | 1.806  | 2.721  | 2.585  | 2.728  | 1.534  | 1.722  | 3.148  | **2.600**   |
>
> Table 4. The Welch’s t-tests between PharmaVQA and the simply augmenting method.
> | Dataset  | T-Statistic | P-Value | Significance (<0.05) |
> |----------|-------------|---------|--------------|
> | BACE     | -2.270      | 0.036   | **Yes**          |
> | BBBP     | -2.676      | 0.015   | **Yes**|
> | ClinTox  | -2.968      | 0.008   | **Yes**|
> | SIDER    | -2.294      | 0.034   | **Yes**|
> | Estrogen | -3.097      | 0.006   | **Yes**|
> | MetStab  | -2.445      | 0.025   | **Yes**|
> | Tox21    | -2.289      | 0.034   | **Yes**|
> | ToxCast  | -1.785      | 0.091   | **No**|
> | Lipo     | 3.375       | 0.003   | **Yes**|
> | Esol     | 2.885       | 0.010   | **Yes**|
> | Freesolv | 3.707       | 0.002   | **Yes**|
>
> Finally, we hope this explanation meet your expectation. Thank you again for your insightful comments, which have been invaluable in refining our work.
>
> ## Response to W6 concerns:
> We would like to express our wholehearful appreciate for your thoughtful feedback and provide us constructive suggestions. We have applied Welch's t-test in the significance analysis experiments. Your feedback has made a significant contribution to improving the quality of my paper.

---

### Official Review · Reviewer_bC5S · 2024-11-08

**Soundness:** 3
**Presentation:** 3
**Contribution:** 2
**Rating:** 6
**Confidence:** 3

**Summary:**

The paper introduces PharmaVQA, an innovative deep learning model designed to enhance molecular representation learning in drug discovery by leveraging pharmacophore information. By integrating a Visual Question Answering (VQA) framework, PharmaVQA effectively retrieves and processes pharmacophore-related data from molecule databases, generating enriched molecular representations that facilitate more accurate drug candidate screening and lead compound optimization. Demonstrated through extensive testing on 46 benchmark datasets and validated on an FDA-approved molecule dataset, PharmaVQA shows superior performance in predicting molecular properties and drug-target interactions, with many of its top predictions confirmed as potential ligands in real-world studies. This model represents a significant advancement in AI-assisted drug discovery, offering a powerful tool to accelerate the development of new drugs across various therapeutic areas.

**Strengths:**

The paper is easy to follow and well-organized, and the idea is quite straightforward. The paper presents PharmaVQA, a novel deep learning model aimed at improving molecular representation learning in drug discovery by utilizing pharmacophore information. By incorporating a Visual Question Answering (VQA) framework, PharmaVQA efficiently extracts and analyzes pharmacophore data from molecular databases, enhancing the representation of drug-like molecules.

The experimental studies are intensive and thorough. Extensive testing on 46 benchmark datasets and validation using an FDA-approved molecule dataset show that PharmaVQA excels in predicting molecular properties and drug-target interactions. Many of its top predictions have been experimentally verified as potential ligands in real-world studies. This model marks a significant step forward in AI-assisted drug discovery, providing a robust tool to speed up the development of new drugs across multiple therapeutic areas.

**Weaknesses:**

The paper does not evaluate the performance on out-of-distribution data. The retrieval-based method requires the test and training data follows the same distribution. However, in drug discovery scenario, the drug molecular distribution varies greatly.

The performance improvement looks marginal.

The test setup for drug-target binding is not fair. We should evaluate the test set where both drug and protein are unseen in training set.

**Questions:**

Please see the weakness part.

---

> ### Author Response · Authors · 2024-11-21
> **Author Response (1/1)**
>
> # Response to Q1: lacks evaluation on out-of-distribution data
> Thank you for your insightful feedback. We would like to clarify the conceptual and methodological nuances of our framework in response to your comment about performance on out-of-distribution data and the limitations of retrieval-based methods.
>
> First, while we use the term "retrieval-augmented," our approach does not conform to traditional retrieval methods that necessitate a shared distribution between training and testing data. Our framework is fundamentally different and does not depend on the assumption of data distribution overlap. Instead, it leverages a visual-question-answering (VQA) mechanism to extract and enhance knowledge directly from the molecules themselves and their textual descriptions.
>
> Our method aims to capture generalized pharmacophore knowledge that is independent of specific molecular distributions. This is achieved through context-aware question-answering, focusing on relationships and features, such as pharmacophore patterns, which are inherently more transferable across diverse molecular structures. By doing so, our approach enhances the model's capacity for generalization to novel distributions, which is a critical aspect of drug discovery where molecular variability is vast and unpredictable.
>
> For Please refer to General Response for further discussion.
>
> # Response to Q2: performance improvement looks marginal
> Thank you for your feedback. We understand your concern about the performance improvement appearing marginal, and we would like to provide further clarification.
>
> Compared to the the state-of-the-art models, our approach demonstrates positive enhancements across certain tasks. For example, in classification tasks such as the BACE dataset, our model demonstrates an approximate 2% improvement compared to the top results offered by KPGT. In regression tasks, such as the Lipo and Freesolv datasets, our model shows improvements of roughly 2% and 9%, respectively. These gains clearly demonstrate integrating pharmacophore-related knowledge, which enhances the model's predictive accuracy and contributes positively to its performance.
>
> Furthermore, we recognize that some datasets, such as SIDER, which involves a multi-label classification task across 27 categories, pose additional challenges that inherently limit the margin for performance enhancement. Despite this, PharmaVQA still demonstrates performance and shows room for further enhancement.
>
> In summary, while the absolute gains might seem limited, they are meaningful within the context of this research field and emphasize the value added by our approach.
>
> # Response to Q3: Unfair comparison on drug-target binding
> Thank you for your feedback regarding the fairness of the drug-target binding test setup. We understand your concern and would like to further explain our experimental design and setup.
>
> We used public datasets, including those from Karimi et al. [１] , Gao et al. [2] and Chen et al. [3], with model results derived from [4]. Importantly, our comparative experiments adhered to the established training and testing split strategies employed in prior research. This ensured that our evaluation approach was aligned with existing benchmarks, making our results directly comparable, fair, and reproducible.
>
> # Reference
> [１].Karimi, M., Wu, D., Wang, Z., & Shen, Y. (2019). DeepAffinity: interpretable deep learning of compound–protein affinity through unified recurrent and convolutional neural networks. Bioinformatics, 35(18), 3329-3338.
>
> [２].Gao, K. Y., Fokoue, A., Luo, H., Iyengar, A., Dey, S., & Zhang, P. (2018, July). Interpretable drug target prediction using deep neural representation. In IJCAI (Vol. 2018, pp. 3371-3377).
>
> [３].Chen, L., Tan, X., Wang, D., Zhong, F., Liu, X., Yang, T., ... & Zheng, M. (2020). TransformerCPI: improving compound–protein interaction prediction by sequence-based deep learning with self-attention mechanism and label reversal experiments. Bioinformatics, 36(16), 4406-4414.
>
> [４].Song, N., Dong, R., Pu, Y., Wang, E., Xu, J., & Guo, F. (2023). PMF-CPI: assessing drug selectivity with a pretrained multi-functional model for compound–protein interactions. Journal of Cheminformatics, 15 (1), 97.

---

### Author Response · Authors · 2024-11-21
**Manuscript Changes**

## Manuscript Changes
All revisions have been highlighted in the updated version of the manuscript. Specifically, sentences have been added to Section 5.2 to address the design of pharmacophore-related questions and to include an ablation study based the feedback provided by Reviewer zSST.

---

> ### Author Response · Authors · 2024-11-21
> **General Response (1/1)**
>
> For Reviewer bC5S：retrieval-based method need same distribution data.
>
> For Reviewer y4UR：Different from traditional "retrieval-augmented" method.
>
> For Reviewer zSST：clarify this usage of "retrieval-augmented" method.
>
> We appreciate the reviewers’ feedback regarding the use of the term "retrieval-augmented" and would like to offer clarification on our approach.
>
> In our work, the term "retrieval-augmented" focuses on constructing pharmacophore-related prompts through a visual-question-answering (VQA) process, allowing the model to extract key pharmacophore information and enhance molecular representations.
> As well known, retrieval is essentially an knowledge extraction process. In our paper, the key contribution of the VQA framework with retrieval augmentation lies in extracting and integrating pharmacophore-related knowledge, which is then injected into the molecular representation to improve model performance. Enhancing the performance of our model by retrieving relevant information. However, unlike traditional RAG frameworks that retrieve documents from large external corpora, our retrieval mechanism focuses on extracting "retrieved" knowledge from structured, domain-specific pharmacophore information.
>
> Although our retrieval method differs from traditional approaches in terms of the knowledge sources, the purpose remains the same: to enhance the model's capabilities by providing more relevant domain knowledge, ultimately leading to more accurate and meaningful results.

---

### Meta-Review · Area_Chair_cPUW · 2024-12-20

**Metareview:**

The paper introduces a deep learning-based model called PharmaVQA for retrieving pharmacophore-related information directly from molecule databases, allowing for a more targeted understanding of drug-like molecules. Through the use of the Visual Question Answering (VQA) framework, PharmaVQA captures pharmacophore data, generates knowledge prompts, and enriches molecular representations. Experiments on 46 benchmark datasets show good performance. This is an AI for Science (drug discovery) submission. After the rebuttal stage, reviewers still kept some unsolved concerns, such as marginal improvements in experimental results and the unfair test setup for drug-target binding. For de novo drug discovery, evaluating the test set where both drug and protein are unseen in the training set is important.

**Additional Comments On Reviewer Discussion:**

There are detailed discussions between the reviewers and authors. The authors have not completely addressed all the questions raised by reviewers. For example, for Q3 of Reviewer bC5S, there are some more papers that DO test unseen drugs and unseen proteins in the experiments. Most reviewers maintain their scores. During the discussion phase, no reviewer championed the paper.

---

### Decision · Program_Chairs · 2025-01-22

Reject